# Measuring compound eye optics with microscope and microCT images

John Paul Currea [1✉], Yash Sondhi [2,3], Akito Y. Kawahara [3] & Jamie Theobald [2✉]

With a great variety of shapes and sizes, compound eye morphologies give insight into visual ecology, development, and evolution, and inspire novel engineering. In contrast to our own *camera-type* eyes, compound eyes reveal their resolution, sensitivity, and field of view externally, provided they have spherical curvature and orthogonal ommatidia. Non-spherical compound eyes with skewed ommatidia require measuring internal structures, such as with MicroCT (µCT). Thus far, there is no efficient tool to characterize compound eye optics, from either 2D or 3D data, automatically. Here we present two open-source programs: (1) the ommatidia detecting algorithm (ODA), which measures ommatidia count and diameter in 2D images, and (2) a µCT pipeline (ODA-3D), which calculates anatomical acuity, sensitivity, and field of view across the eye by applying the ODA to 3D data. We validate these algorithms on images, images of replicas, and µCT eye scans from ants, fruit flies, moths, and a bee.

[1] Integrative Biology and Physiology, UCLA, Los Angeles, CA 90095, USA. [2] Department of Biological Sciences, Florida International University, Miami, FL 33199, USA. [3] McGuire Center for Lepidoptera and Biodiversity, Florida Museum of Natural History, University of Florida, Gainesville, FL 32611, USA. ✉email: pablocurrea@ucla.edu; theobald@fiu.edu

Arthropods, with about 1.3 million described species, represent roughly 80% of all known animal species[1]. They range vastly in size, with body lengths from the 85 μm ectoparasitic crustacean *Tantulacus dieteri*[2] to the over 50 cm green rock lobster, *Sagmariasus verreauxi*[3,4], and lifestyle, with activity in nearly every ecological niche. Likewise, arthropods wield an array of eye architectures, most commonly compound eyes[5,6]. The arthropod compound eye has been a model for understanding cellular fate and neural development[7–9] and comparing eyes across species can reveal underlying selective pressures driving eye evolution[10–14]. It has further sparked innovations in artificial eyes, computer vision, and nano-technology, including the development of anti-reflective coatings that imitate the graded refractive indices of some insect eyes and three-dimensional eye ultrastructure designs that enhance solar panel light absorption[15–17].

Eye morphology is fundamental to how animals see because it sets physical limitations on the capacity to form images[6]. Depending on the light intensity, spectral characteristics, and image motion, some optimal eye architecture will maximize the ability to gather image information[18–20]. Because of the critical role of eye morphology in understanding visual ecology,

development, and evolution, we offer a program to accurately and automatically characterize compound eye optics.

In contrast to the camera-type eyes we possess, compound eyes are made up of multiple, repeated optical elements that are externally visible. These *ommatidia* individually direct light onto photoreceptors. Contrary to popular belief, they generally do not produce a myriad of tiny images on the retina, but average into the functional pixels of the transduced image. The number of ommatidia therefore determines the total number of images an eye can form, or its spatial information capacity. Ommatidia can be counted in micrographs, ranging from about 20 in the fairyfly *Kikiki huna* (body length = 158 μm)[21,22] to over 30,000 in large dragonflies[5]. Compound eyes further divide into two structural groups: apposition eyes, in which pigment cells between ommatidia restrict incoming light to a single rhabdom, such that lens size limits optical sensitivity (Fig. 1a), and superposition eyes, in which light travels through a clear zone that allows many facets to contribute to each point (Fig. 1b), thereby multiplying the final sensitivity.

Previous studies have relied on painstaking manual counts and estimates to describe compound eye structure. Fortunately, ommatidia count and diameter estimation from 2D and 3D eye images can be automated. Although several algorithms and software plugins have been proposed, they currently require user input for each image, and frequently underestimate ommatidia counts[23], overestimate ommatidial diameter[24], or were not validated against manual measurements or measurements in the literature[25–27]. They do work in limited cases with a few hundred clearly separated ommatidia, but have not been tested on multiple species, over a substantial range of eye sizes, or with different media. Since the pre-print of this manuscript, a method has been proposed for processing CT data but it relies on access to proprietary MATLAB software[28].

Here, we offer two open-source programs written in Python to characterize compound eyes: (1) the ommatidia detecting algorithm (ODA), which identifies individual facets (the outward visible portion of the ommatidia) in 2D images, and (2) a multistage μCT pipeline (ODA-3D) which applies the ODA to segment ommatidia components and characterize the visual field. We test the reliability and validity of this technique on single images of 5 eye molds of 4 different ant species ranging in size, light micrographs of 29 fruit flies (*Drosophila melanogaster*), scanning electron micrographs (SEMs) of fruit flies (*D. melanogaster* and *D. mauritania*) and processed images of μCT scans of one fruit fly (*D. mauritania*), 2 moths (*Manduca sexta* and *Deilephila elpenor*) and one bee (*Apis mellifera*).

For spherical eyes, the lens diameter measurements provided by the ODA can be divided by measurements of eye radius (using the luminous pseudopupil technique, for instance) to measure the angular separation of ommatidia, called the interommatidial (IO) angle (Fig. 1 $\Delta\varphi$). The inverse of this angle limits spatial acuity[18–20]. High spatial acuity affords many behaviors, such as prey, predator, and mate detection, and perceiving small changes in self-motion[6,18]. For spherical eyes, the IO angle is approximately: $\Delta\varphi = D/R$, where $D$ is the ommatidial lens diameter and $R$ is the radius of curvature, assuming the axes of all ommatidia converge to a central point. Fortunately, many compound eyes closely approximate the spherical model. Smaller compound eyes are often spherical and homogenous because photon noise and diffraction constrain the range of viable IO angles and ommatidial sizes[20]. Likewise, superposition eyes are often roughly spherical because they must optically combine light from many ommatidia, which constrains their heterogeneity (Fig. 1b)[6,29].

Eyes with the longitudinal axes of ommatidia askew to the eye surface are not well approximated by a spherical model. Skewed ommatidia can improve acuity at the expense of field of view

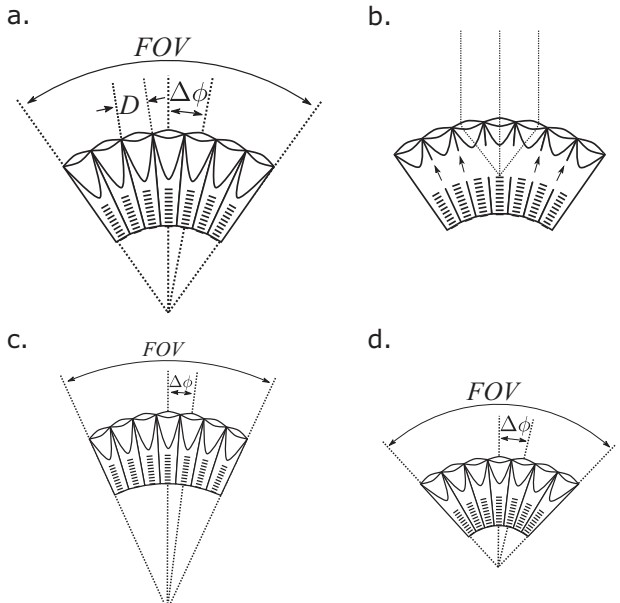

**Fig. 1 Geometric tradeoffs between lens diameter (*D*), interommatidial angle (*Δφ*), and field of view (*FOV*).** Diagrams of apposition and superposition eyes demonstrating the geometric tradeoffs between *D*, *Δφ*, and *FOV* for spherical (a. and b.) and non-spherical eyes (c. and d.). **a** In spherical apposition eyes, *D* directly determines sensitivity while *Δφ* inversely determines acuity. **b** In superposition eyes, migrating pigment (indicated by the arrows) allows the ommatidia to share light, increasing the eye's sensitivity. As a result, these eyes generally adhere to a spherical design. **c-d** In nonspherical eyes, the intersection of ommatidial axes differs from the center of curvature, with ommatidial axes askew from the surface of the eye. Consequently, *FOV* and *Δφ* are not externally measurable and the effect of *D* on sensitivity is reduced by greater angles of skewness. **c.** When the distance to the intersection is greater than the radius of curvature, *FOV* and *Δφ* decrease, increasing average spatial acuity by directing more ommatidia over a smaller total angle. **d** Inversely, when the distance to the intersection is less than the radius of curvature, *FOV* and *Δφ* increase, decreasing average spatial acuity by directing fewer ommatidia over a smaller total angle. In both cases, optical sensitivity is lost because skewness reduces the effective aperture of the ommatidia.

(FOV) by pointing more ommatidia onto a small visual field (Fig. 1c), or increase FOV at the expense of acuity by spreading a few ommatidia over a large visual field (Fig. 1d), but in both cases sacrifice sensitivity by reducing the effective aperture as a function of the skewness angle and refraction[30]. For more information on the optical consequences of ommatidial skewness, see Stavenga (1979)[30]. μCT allows calculating visual parameters for non-spherical compound eyes, measuring anatomical IO angles at high spatial resolution, and segmenting different tissues, such as visual neuropils, within the same dataset. It is a quickly growing technique for comparative morphology[31,32], used on arthropods to study muscles[33], brains[34], ocelli[35], and eyes[13,36–38]. Our second proposed method, ODA-3D, segments individual corneal lenses or crystalline cones from a μCT image stack (using the ODA) to measure the size, orientation, and spatial distribution of ommatidia. We validate it on μCT scans of approximately spherical fruit fly and moth eyes and a nonspherical honeybee eye, and further demonstrate how ODA-3D can detect oval eye features, measure regional changes in skewness, spatial acuity, and sensitivity, and project onto world-referenced coordinates to accurately measure FOV.

Because there are so many arthropod species and compound eye morphologies, it is challenging but valuable to characterize them in a meaningful, fast manner. Here we demonstrate the operational range of two programs to automate this task as a function of image resolution and contrast and benchmark its performance against estimates of the time needed to take comparable measurements by hand. Overall, our proposed methods minimize the substantial labor that is typically required in characterizing optical performance in compound eyes and therefore facilitate understanding their role in vision.

## Results

**Microscope images**. We tested the ODA on 4 sets of images: 1) light micrographs of the flattened eye molds of 5 ants of 4 different species (Fig. 2a), 2) light micrograph focus stacks of 5 *D. melanogaster* specimens, 3) SEMs of 5 different *D. melanogaster* specimens, and 4) SEMs of 5 *D. mauritiana* specimens. To assess the performance limitations of the ODA, we applied it to each image after programmatic degradation of image resolution and contrast (see methods for more detail). We report the runtime, output number, and diameter of ommatidia for each image and degradation level to compare with manual measurements.

Images at full resolution and contrast produced the most accurate automated measurements of ommatidial count (automated/manual = 94 ± 13%; mean ± standard deviation) and diameter (86 ± 7%). Among media types, the ant eye replicas were closest to manual measurements (count: 99 ± 3%; diameter: 93 ± 3%), followed by the *D. melanogaster* micrographs (c: 105 ± 12%; d: 82 ± 9%), the *D. mauritiana* SEMs (c: 88 ± 2%; d: 82 ± 5%), and the *D. melanogaster* SEMs (c: 81 ± 6%; d: 86 ± 4%). The ant eye replicas likely performed best because they physically unwrap the eye surface, reducing the distortion due to the eye curvature, and have a sharp, high contrast spot at the center of each ommatidium unlike the smooth, low contrast SEMs.

Reducing spatial resolution had a predictable effect on ODA output (Fig. 2b, left column). At degraded resolutions, measurements of ommatidia count and diameter did not change substantially as long as the pixel resolution was sufficiently above the Nyquist limit set by the ommatidial diameter. Based on the Nyquist criterion[39], the image must have at least two pixels for every ommatidial diameter to properly resolve the ommatidial lattice. To characterize this threshold resolution, we measured the lowest image resolution resulting in a measurement greater than

50% of the maximum relative ommatidial count and less than 50% of the maximum relative lens diameter per subject. Across all media, the threshold resolution for ommatidial count was 4.2 ± 1.9 pixels per diameter (px/D) and for lens diameter was 4.5 ± 2.3 px/D, close to the theoretical Nyquist limit of 2 px/D. The *D. melanogaster* SEMs performed better than the other media (2.9 ± 0.1 px/D), followed closely by the ant eye replicas (3.2 ± 1.0 px/D). In terms of lens diameter, the ant eye replicas performed just above the theoretical limit (2.1 ± 0.7 px/D). The runtime also followed a predictable trend where higher resolutions resulted in longer runtimes. However, not even the highest resolutions took longer than 8 s, which is a substantial improvement over the time needed to count the ommatidia by hand (which is on the order of 10 minutes to an hour depending on the number of ommatidia).

At lower contrasts, measurements of ommatidia count and diameter changed substantially for the SEMs and 2 of the ant eye replicas at the lower contrasts (Fig. 2b, right column). We characterized the threshold contrasts in the same way as the threshold resolutions above. Across all media, the threshold contrast for ommatidial count was 0.01 ± 0.01 root mean square (RMS) and for lens diameter was 0.01 ± 0.003 RMS. Although the threshold contrast was roughly the same for all media, the SEMs were particularly susceptible to reduced contrast, dropping from a relative mean ommatidial count of 81% to 2% for the *D. melanogaster* and 88% to 0% for the *D. mauritiana* SEMs at the lowest contrast as opposed to reductions from 105% to 99% for the *D. melanogaster* micrographs and 99% to 69% for the ant eye replicas. As for mean ommatidial diameter, the SEMs again showed the most sensitivity to reduced contrast, increasing from 86% to 234% for the *D. melanogaster* and 82% to 93% for the *D. mauritiana* SEMs at the lowest contrast as opposed to increases from 81.7% to 82.3% for the *D. melanogaster* micrographs and 92% to 93% for the ant eye replicas. This may be due to the already low contrast of SEMs, whereas the other images have high contrast reflections of a light source near the center of each lens. Even so, the ODA performed well over a nearly tenfold range of contrast reductions for all media. The ODA was successful on images that were such low contrast that we struggled to see individual ommatidia ourselves. Further, contrast had no clear effect on runtime, which again never exceeded 8 s. The runtime was roughly the same across all ant eye replicas—ranging from 153 to 2626 ommatidia—suggesting that runtime is not substantially affected by ommatidia count. The ODA should therefore work quickly on images with a resolution exceeding the Nyquist limit and a reasonable contrast. A resolution of about 10 px/D would provide equivalent results to higher resolutions and significantly reduce the program's runtime.

We also tested microscope images of the eyes of 29 vinegar flies (*D. melanogaster*; Fig. 2c and d) to determine ODA performance on comparisons within a species. Visual inspection of preliminary results found that using just the first 2 fundamental frequencies resulted in substantially fewer false positives than checking for all 3. These false positives—likely responsible for the overestimates above—may be due to noise induced by the curvature of the eye and the quality of the image. Automated counts and diameters shared strong and significant correlations with manual measurements (counts: r = .81, df = 27, p ≪ .001; diameters: r = .76, df = 27, p ≪ .001), and automated counts were 100 ± 4% and diameters 95 ± 2% of those taken by hand offering precise and relatively accurate estimations.

**μCT**. We tested ODA-3D on eye scans of a fruit fly (*D. mauritiana*) collected by Maike Kittelmann and used with her permission, two moth species (*M. sexta* and *D. elpenor*) that we

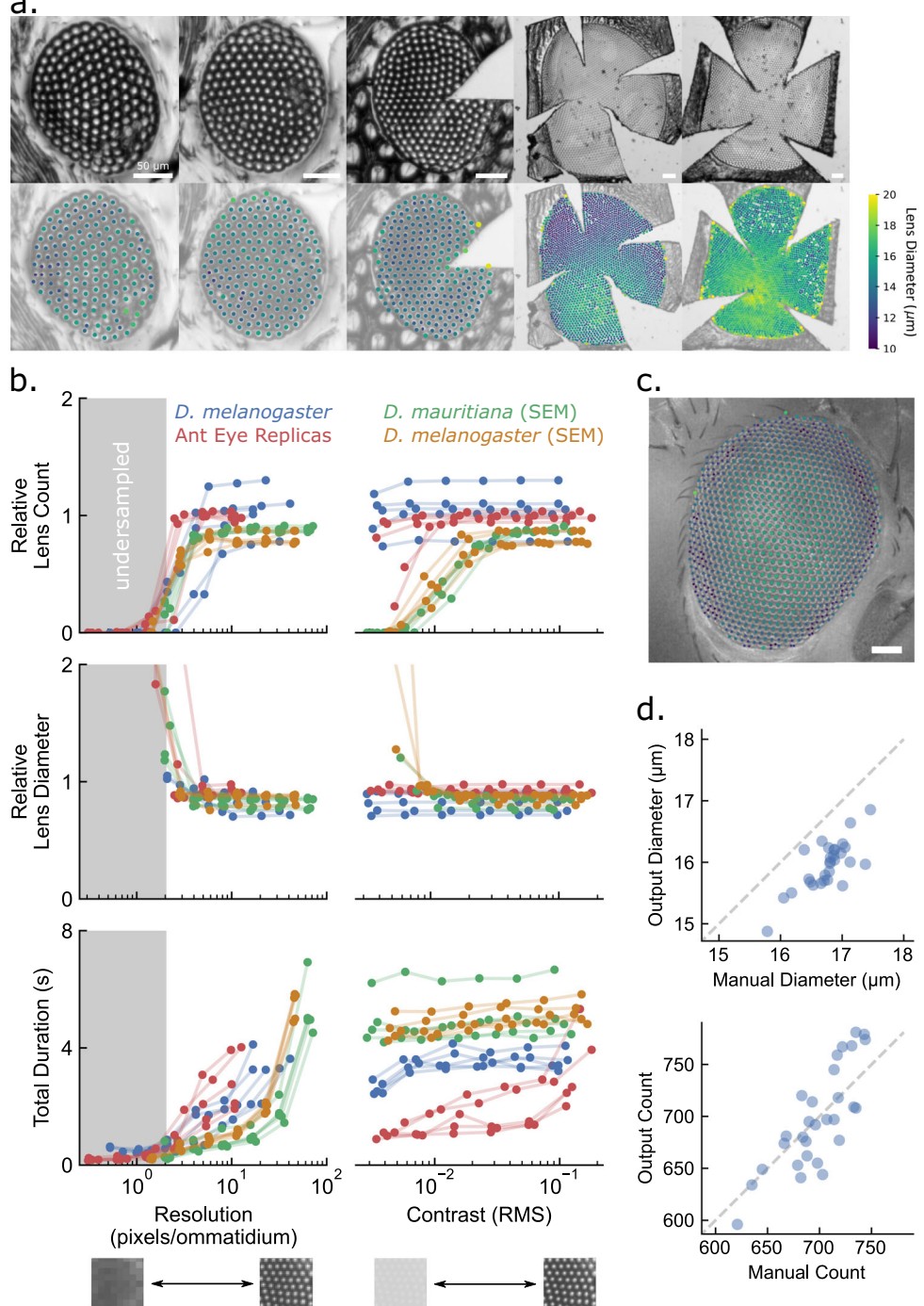

**Fig. 2 The ODA successfully approximated ommatidial counts and diameters when compared to manual measurements. a** Micrographs of ant eye molds (top) and their corresponding ODA results (bottom) with approximate lens centers colored according to their diameter. When applied to 5 ant eye molds of 4 species ranging in overall size and lattice regularity, the automated counts were 99% and the diameters were 93% of those measured by hand. For each image, the program missed relatively few ommatidia and the lens diameter measurements were successful even when they varied substantially within an image (as in #4 from the left). Species from left to right: *Notoncus ectatommoides*, *Notoncus ectatommoides*, *Rhytidoponera inornata*, *Myrmecia nigrocincta*, and *Myrmecia tarsata*. Scale bars are 50 μm. **b** Benchmark performance of ODA on micrographs of diminished spatial resolution (left) and contrast (right). We present 3 performance metrics as a function of resolution and contrast: relative lens count equal to the ratio of automatic to manual ommatidia counts (top); relative lens diameter equal to the ratio of automatic to manually measured lens diameters (middle); and the total duration or runtime of the ODA (bottom). **c** An example fruit fly eye (*D. melanogaster*) micrograph with the automated ommatidia centers superimposed as points colored according to the measured diameter using the same colormap as in a. **d** A comparison of automated and manual measurements of lens diameter and count for 29 microscope images of fruit fly eyes from the same species (*D. melanogaster*). Automated counts were 100% and the diameters were 95% of those measured by hand, with correlations of .81 and .76. Again, there were relatively low rates of false positives and negatives.

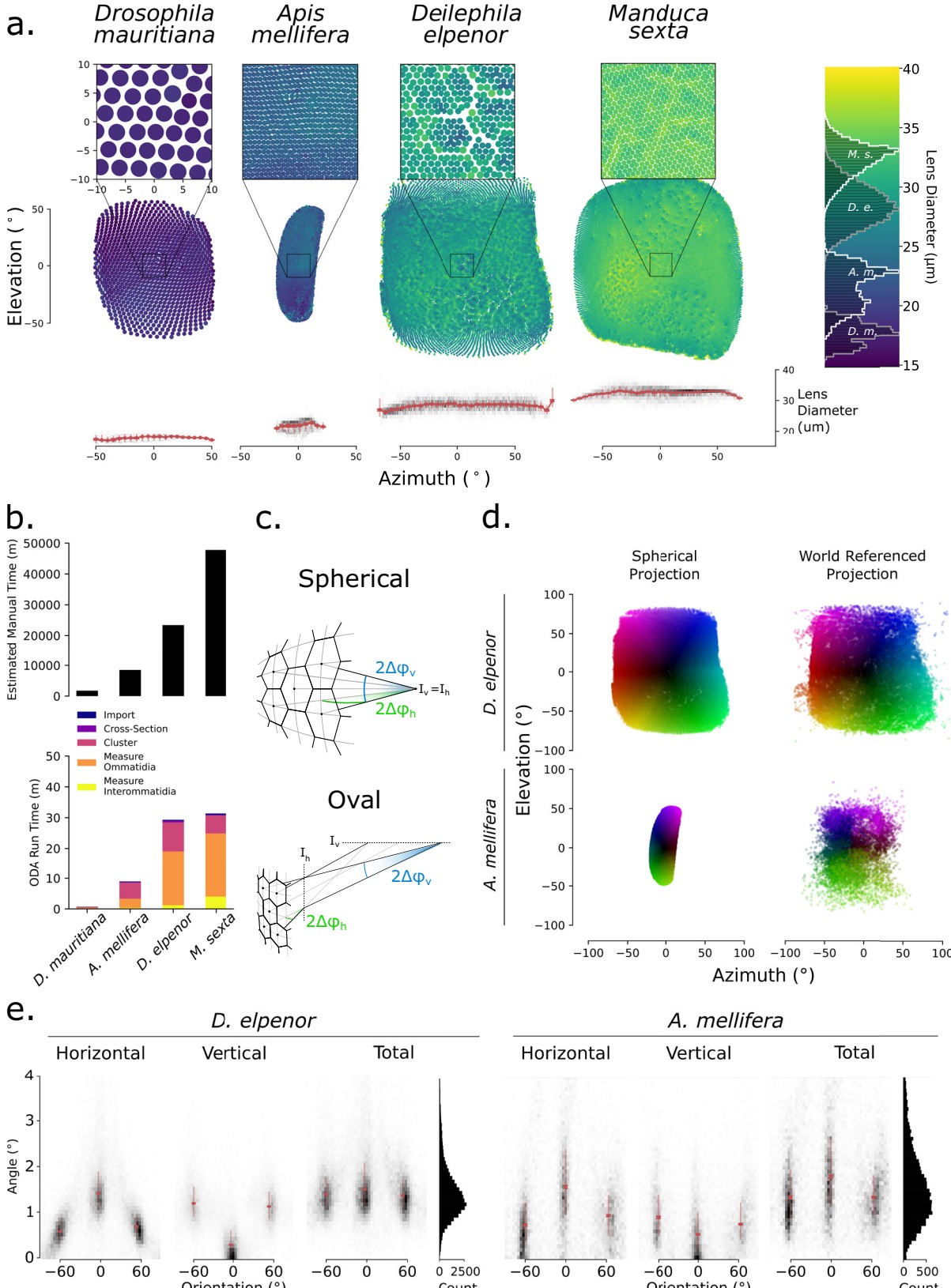

collected, and one honeybee (*A. mellifera*) used in Taylor et al.[37] (Fig. 3a). The measurements of lens diameter, skewness, skewness-adjusted lens diameter, and spherical, anatomical, and modeled IO angles for each specimen can be found in Table 1. We additionally tested methods for non-spherical eyes on the

honeybee scan, including features of an oval eye and projecting onto world-referenced coordinates.

Automated ommatidia counts were 100.4% of the manual count on the same *D. mauritiana* scan, 85% of the density-based count of the same *A. mellifera* scan[28,37], 112% of measurements

**Fig. 3 Comparing visual fields based on the spherical approximation. a** ODA-3D allows us to map the visual fields of our four specimens (*Drosophila mauritiana, Apis mellifera, Deilephila elpenor,* and *Manduca sexta*). Using their spherical projection, we map the azimuth (along the lateral-medial axis) and elevation (along the ventral-dorsal axis) angles of ommatidia and represent lens diameter with color. Note that the bee eye was flipped horizontally from the images in Fig. 6. The colorbar to the right indicates lens diameter and shows diameter histograms for each species in white and gray. Insets zoom in on a 20° x 20° region in the center of each eye, showing ommatidial lattices in more detail. Note that the spherical projection of the bee eye underestimates its visual field as explained in the text. **b** The estimated time to segment each stack, based on the 2 min per cone estimate from Tichit et al. (2022), is plotted above the ODA-3D runtime. Note the 1000-fold difference in the y-axes illustrating that the ODA-3D runs at about 1000 times faster than the estimated time to segment the crystalline cones manually. **c** Adapted from Stavenga, 1979. Bee eyes are approximately oval, and, unlike spherical eyes, have different intersection points for horizontal ($I_h$) and vertical ($I_v$) ommatidial pairs corresponding to different IO angle components horizontally ($\Delta\varphi_h$) and vertically ($\Delta\varphi_v$). For oval eyes, the horizontal component of the IO angle of horizontal pairs (orientation=0°) is $2\Delta\varphi_h$ while the vertical component is ~0° and the horizontal component of diagonal pairs (orientation = ±60°) is $\Delta\varphi_h$ while the vertical component is $\Delta\varphi_v$. **d** To demonstrate the difference between spherical and oval eye visual fields, we plot the spherical and world referenced projections of the ommatidial axes for the moth (*D. elpenor*) and bee (*A. mellifera*) eyes. Notice that there is little difference for the spherical moth eye but a substantial difference horizontally for the bee eye. **e** The vertical and horizontal subtended angles of ommatidial pairs with respect to their orientation also demonstrate differences between the spherical moth (*D. elpenor*) eye and the oval bee (*A. mellifera*) eye. The pair orientations form trimodal distributions with means close to ±60° (diagonal pairs) and 0° (horizontal pairs) for both species, but the distributions are more uniform for the moth eye. Instead, the bee eye has significantly larger horizontal than vertical pairs and each pair group is more variable, resulting in a larger combined distribution of IO angles. Grayscale heatmaps present 2D histograms and medians (red point), IQRs (half-opacity red line), and 99% C.I.s (full opacity error bars) are plotted for each orientation ±15°.

**Table 1 Optical parameters for the four species.**

|  | *D. mauritiana* | *A. mellifera* | *D. elpenor* | *M. sexta* |
|---|---|---|---|---|
| Count | 979 | 4725 | 13004 | 26552 |
|  | 975*, 957 ± 22.89[42], 1018[43] | 5440 [37]* | 11508[40] | 27000[41], 25641[40] |
| Lens Diameter (µm) | 17.69 ± 0.92 | 21.79 ± 1.75 | 28.47 ± 1.86 | 32.32 ± 1.53 |
|  | 15.37 ± 1.89[42], 18.55 ± 1.07[43] | 22.61 ± 1.96 [37]* | 28[40], 29[44] | 30[41], 31[40], 34[44] |
| Skewness (°) | 4.55 (7.02) | 12.81 (11.73) | 5.92 (4.28) | 5.79 (4.51) |
|  |  | ≤ 50[45] |  |  |
| Adjusted Lens Diameter (µm) | 17.40 ± 1.15 | 20.71 ± 2.33 | 28.18 ± 1.93 | 32.04 ± 1.65 |
| Spherical IO Angle (°) | 3.31 (0.26) | 0.87 (0.11) | 1.17 (0.10) | 0.83 (0.05) |
| Anatomical IO Angle (°) | 3.87 (1.64) | 1.33 (1.07) | 1.42 (0.69) | 1.04 (0.52) |
| Modeled IO Angle (°) | 3.85 | 1.05 | 1.34 | 0.93 |
|  |  | 1.62 (0.61) [37]* | 1.12[40], 1.31[44] | 0.91[40], 0.96[44] |

Values are mean ± s.d except for angular measurements, which show median (IQR). Comparable measurements from the literature or measured manually are underscored below corresponding values, with an asterisk indicating manual measurements of the same dataset.

of a different *D. elpenor* specimen[40], and 98% and 104% of measurements of different *M. sexta* specimens[40,41]. These ommatidia had diameters (Fig. 3b) that were consistent with measurements in the literature: 95–115% for *D. mauritiana*[42,43], 96% for *A. mellifera* despite missing ~15% of ommatidia[37], 99 – 103% for *D. elpenor*[40,44], and 96–108 % for *M. sexta*[40,41,44]. We also profiled the runtime of the ODA-3D on the 4 specimens in comparison to estimates of how long it would take to extract the clusters manually. Estimates of manual time are based on Tichit et al.[28], which recorded about 1.8 min to manually segment each of 100 crystalline cones in the *A. mellifera* stack. Assuming this same rate for all of our specimens, our program offered comparable results 883 to 2425 times faster than manual segmentation (note the 1000-fold difference in the y-axes). Moreover, these estimates only account for the segmentation stages of the procedure, omitting the time needed for taking ommatidial and interommatidial measurements which represented the majority of the ODA runtime for the two moth scans.

The fly eye and both moth eyes showed ommatidial axes with minor skew from spherical alignment. The bee, however, showed axes with substantial skew, consistent with other anatomical measurements[45]. Skew geometrically reduces effective lens diameter by the cosine of skew angle, and further still by refraction of the image space[30]. The adjustment without accounting for refraction produced marginal reductions for the spherical eyes: 0.3% in *D. mauritiana*, 0.5% in *D. elpenor*, and 0.5% in *M.sexta*, but more substantial reduction for the oval eye: 2.5% in *A. mellifera*. The spherical approximations for IO angle

were closer to anatomical IO angle in moth eyes than the honeybee eye: 86% in *D. mauritiana*, 82% in *D.elpenor*, and 80% for *M. sexta* versus 65% in *A. mellifera*. Aside from the spherical approximation for *A. mellifera*, IO angles were consistent with previous measurements in the literature: 54% for the spherical and 82% for the anatomical approximation in *A. mellifera*[37], 89–97% for the spherical and 108–127% for the anatomical approximation in *D. elpenor*[40,44], and 86–91% for the spherical and 108–114% for the anatomical approximation in *M. sexta*[40,44]. To our knowledge, previous measurements of *D. mauritiana* IO angles are unavailable in the literature, but our measurement of 3.9° is in the range of *Drosophila melanogaster*, which is on average 4.5° and as low 3.4°[46].

**Nonspherical properties.** In bees, spherical coordinates largely accounted for vertical axis curvature, but vastly underestimated horizontal curvature. To better characterize their visual field, we projected the ommatidial axes onto a sphere outside of the eye, like the world-referenced projection of[37]. We used the center from step B of ODA-3D, which is near the center of the head and chose a radius of 10 cm based on visual fixation behavior[47]. As opposed to *D. elpenor*, which had very similar spherical and world-referenced projections, the *A. mellifera* spherical projection largely underestimated the vertical FOV as 54° and horizontal FOV as 21° (Fig. 3d). The world-referenced visual field, subtending about 110° horizontally and 126° vertically, was closer to previous measurements of 140° horizontally and 162° vertically

based on vectors orthogonal to the eye surface[37]. This discrepancy was likely due to ODA-3D errors during the segmentation of highly skewed ommatidia in the periphery (for example, see the horizontal slice insets in Fig. 6). Glial pigment cell interactions between the crystalline cones or deformation of the soft cells could also have caused the error. For precisely measuring the FOV of eyes with highly skewed ommatidia, we recommend a process that uses subvolume unfolding like Tichit et al.[28].

Nonetheless, by accounting for some ommatidial skewness, ODA-3D allows us to compare the structural properties of spherical and non-spherical eyes. The bee eye is an oval eye, with ommatidial axes intersecting at different points for horizontal and vertical IO pairs ($I_h \neq I_v$ in Fig. 3c). Anatomical IO angles are therefore separable into independent horizontal and vertical components ($\Delta\varphi_h$ and $\Delta\varphi_v$ in Fig. 3c). For ommatidia arranged in a regular hexagonal lattice, the orientations of IO pairs should fall into 3 modes separated by 60°; 2) the horizontal angle for horizontal IO pairs is $2\Delta\varphi_h$ while the horizontal angle for diagonal IO pairs is $\Delta\varphi_h$; 3) the vertical angle is 0 for horizontal pairs and $\Delta\varphi_v$ for diagonal pairs; and 4) the proportion $\Delta\varphi_v / \Delta\varphi_h$ is approximately $1/\sqrt{3}$[30]. Finally, for an oval eye to follow a regular hexagonal lattice, 5) the vertical radius of curvature, $R_v$, must be 3 times the horizontal radius, $R_h$, whereas a spherical eye requires $R_v = R_h$.

Both eyes are consistent with the criteria of a regular hexagonal lattice. 1) The IO orientations of both eyes follow trimodal distributions with modes separated by about 60° (Fig. 3e). IO pairs within 15° of the three modes were selected to measure horizontal and vertical angles and calculate the horizontal and vertical IO angle components as in Fig. 3c. For the moth-eye, 2) the horizontal angle for horizontal IO pairs, 1.42°, is nearly twice the horizontal angles for diagonal pairs, 0.71° + 0.60° = 1.31°; 3) the vertical angles 0.31°, are close to 0° for horizontal pairs and are nearly equal for diagonal pairs, 1.14° and 1.21°; and 4) the proportion $\Delta\varphi_v / \Delta\varphi_h = 0.40$ is close to $1/\sqrt{3}=0.58$. For the bee eye, 2) the horizontal angle for horizontal IO pairs, 1.59°, is nearly twice the horizontal angle for diagonal pairs, 0.96° + 0.74° = 1.70°; 3) the vertical angles for horizontal pairs, 0.54°, are close to 0° and are nearly equal for diagonal pairs, 0.76° and .92°; and 4) the proportion $\Delta\varphi_v / \Delta\varphi_h = 0.73$ is close to.58.

For the oval eye criterion, we calculated the radius for each mode based on equation (1): $R = D/\Delta\varphi$, where $R$ is the radius of curvature or intersection, $D$ the ommatidial diameter, and $\Delta\varphi$ the IO angle. If the radius of curvature follows an elliptical function of IO orientation, then diagonal pairs should have a radius of curvature ~2.23 times the horizontal radius. In the moth-eye, both diagonal and horizontal pairs had roughly the same radius of intersection ($R_d = 1188$ μm; $R_h = 1079$ μm; $R_d/R_h = 1.10$). In the bee eye, however, the mean diagonal radius ($R_d = 923$ μm) is 1.44 times the mean radius for horizontal pairs ($R_h = 640$ μm), closer to an oval eye.

By combining the horizontal angle of diagonal pairs and half the horizontal angle of horizontal pairs, we approximated the horizontal anatomical IO angle component, $\Delta\varphi_h = 0.82 \pm .81°$ ($N = 11,312$). Again, this is almost certainly an underestimate due to poor horizontal segmentation of crystalline cones in the lateral eye. Previous measurements in the literature, which found $\Delta\varphi_h$ to be 1.1–1.9° across different regions of the eye[48]. By combining the vertical angles of diagonal pairs we approximate $\Delta\varphi_v = .84 \pm 0.83°$ ($N = 7111$) and the total anatomical IO angle ($\sqrt{\Delta\varphi_v{}^2 + \Delta\varphi_h{}^2}$) as $\Delta\varphi = 1.17°$. This is close to the estimate based on the anatomical angle of all IO pairs, $1.33 \pm 1.07°$. Despite underestimated horizontal IO angle components, both estimates are consistent with measurements on the same scan assuming ommatidial axes orthogonal to the eye surface, which found IO angles between 0.9° and 1.7°[37].

## Discussion

Our methods successfully automate the estimation of multiple visual parameters of compound eyes. We tested compound eye images with the ODA, which filters spatial frequencies based on the hexagonal arrangement of most ommatidia and applies a local maximum detector to identify their centers. The ODA calculated ommatidial count and lens diameter from different media (eye molds, microscope images, and μCT scans), taxa (ants, flies, moths, and a bee), sizes (hundreds to tens of thousands of ommatidia), and eye types (apposition, neural superposition, and optical superposition). In all cases, measurements provided by the program matched with manual measurements on the same data, previous measurements in the literature, or both. Ommatidial counts were accurate when compared to previous measurements on the same dataset or in the literature: 95 ± 15% for the datasets in the ODA benchmark, with ant eye replicas performing the best at 99 ± 3%; 100 ± 4% for the *D. melanogaster* micrographs; and 99.6% (range=85–112%) for the 4 CT scans. Ommatidial diameters were also accurate: 86 ± 6% for the datasets in the ODA benchmark, with ant eye replicas performing the best at 93 ± 3%; 95 ± 2% for the *D. melanogaster* micrographs; and 101% (96–105%) for the 4 CT scans.

The ODA-3D, which integrated the ODA into a μCT pipeline, proved successful on scans of one spherical fruit fly eye (*D. mauritania*), two spherical moth eyes (*D. elpenor* and *M. sexta*) and one oval bee eye (*A. mellifera*). In addition to counts and diameters, ODA-3D estimated anatomical IO angles, FOV, and skewness. Skewness angles were insignificant in moth eyes, which generally require approximate sphericity for proper optical superposition. However the oval honeybee eye showed significant skewness angles, implying reduced optical sensitivity, lower resolution and a greater FOV horizontally. Again, estimates were consistent with manual measurements on the same data, previous measurements in the literature, or both. High-resolution 3D data additionally offered world-referenced coordinates and measurements of resolution at different angles along the eye to better characterize the visual field.

The great eye size range between and among invertebrate species[6,10,11,13,18,49] makes compound eyes ideal for studying environmental reaction norms and allometry. Little allometry research deals with compound eyes, and instead favors organs easily measured in one or two dimensions[49–51]. By automating the more tedious tasks of characterizing compound eyes, our programs should help with this challenge. For instance, ODA counts and diameters allow total cell count approximations and correspond to the independent effects of cellular proliferation and growth during eye development. Further, our program facilitates measuring allometry of visual performance, addressing the environmental reaction norms of the anatomical determinants of vision. Further, progress in understanding fruit fly (*D. melanogaster*) eye development[7,9,10,13], makes compound eyes ideal for assessing principles of eye development across different taxa[10,12,14]. And because optics are the first limit to incoming visual information[18,30], they inform electrophysiological and behavioral data to infer intermediate neural processing[6,11,44,46,52–55].

Our programs have some known limitations. Images for the ODA require sufficient resolution to properly detect ommatidia. For regular images, pixel length must be at most half the smallest ommatidial diameter according to the theoretical limit and is closer to .25 or lower in practice. For μCT, if individual crystalline cones cannot be resolved at each layer, they are likely indiscriminable to the ODA. Further, some species, preparations, and scanning procedures, capture better contrast between crystalline cones and other structures while avoiding structural damage to the specimen. For example, *M. sexta* crystalline cones contrasted sharply with the background scan when prefiltered with just the

threshold function. *D. elpenor*, however, had additional noise outside of the eye, and *A. mellifera* had uneven exposure, altering density measurements across locations in the scan and ultimately forcing the omission of some data. This may be an unintentional consequence of the contrast enhancing property of a synchrotron light source[31]. Most importantly, ODA-3D erroneously segmented highly skewed ommatidia in the *A. mellifera* scan, resulting in inaccuracies downstream. This could likely be improved by incorporating nonspherical subvolume unfolding, like in Tichit et al. (2022)[28]. Preservation techniques can cause small deformations in the eye and while ODA-3D still works on minor deformations, scans of highly deformed eyes break certain assumptions about the uniformity of the lattice and cannot be analyzed using this method. We also make certain assumptions in calculating parameters like ommatidial diameter, interommatidial angle that may introduce some inaccuracies at the edges, however, we allow the user to omit data points that are significant outliers.

Ultimately, anatomical measurements cannot replace optical techniques in measuring compound eye optics[30,37]. Light passing through an eye refracts depending on the incident angle and index of refraction[30]. But our approximations used only incident angle, so our measurements of the aperture-diminishing effect of skewness represent lower bounds, and our measurements of IO angles are *anatomical*, not functional IO angles. Though skewness can be somewhat corrected for, nothing can match optical techniques[30,37].

Future work will be needed to understand the limitations of both the ODA and ODA-3D. Because the ODA depends on spatial frequencies corresponding inversely to the ommatidial diameter, an eye with a wide range of diameters or high curvature may not work, and the ODA should be tested on eyes containing acute or sensitive zones, such as the robberfly[54] and lattices with

transitioning arrangements, such as from hexagonal to square in houseflies[56] and male blowflies[57]. Likewise, the ODA-3D should be tested on non-spherical non-oval eyes. While our program appropriately measured anatomical IO angles across much of the honeybee's oval eye and actually corroborated its oval eye properties, it may not work when IO angles change dramatically like in the robberfly[54]. Finally, the ODA-3D should be tested on non-insect arthropods.

Compound eyes are the most common eye type on Earth, found in nearly every ecological habitat and visual environment, and varying widely in size, shape, and architecture. Because they are diverse, ubiquitous, and subject to heavy selection pressure, they are crucial to understanding the evolution of vision. Our programs contribute to this effort and are open source, easy to install, easy to incorporate into custom pipelines, and downloadable as a Python module. By successfully measuring parameters from a wide range of eye shapes and sizes, they should facilitate the study of the development, evolution, and ecology of visual systems in non-model organisms.

## Methods

**Specimens and eye imaging.** Micrographs of glue eye molds or replicas were taken previously on 5 ant specimens from four ant species: two *Notoncus ectatommoides* of the Formicinae subfamily (from Palavalli-Nettimi and Narendra, 2018)[58], a jumper ant (*Myrmecia nigrocincta*) and a bull ant (*M. tarsata*) of the Myrmeciinae subfamily, and *Rhytidoponera inornata* of the Ectatomminae subfamily (from Palavalli-Nettimi et al. (2019))[59]. Micrographs of 29 fruit fly eyes (*D. melanogaster*) were also drawn from Currea et al. (2018)[11] and SEMs of two fruit fly species (*D. melanogaster* and *D. mauritiana*) collected by Maike Kittelmann. For the SEMs, fly heads were removed from the body and placed into Bouin's solution (Sigma Aldrich) overnight at room temperature. Heads were then dehydrated in an ethanol series of 50, 70 and 3×100%. Heads were then critical point dried, mounted onto sticky carbon tabs on 12 mm SEM stubs, sputter coated with 15 nm gold and imaged at 5 kV in a Hitachi S-3400N with secondary electrons.

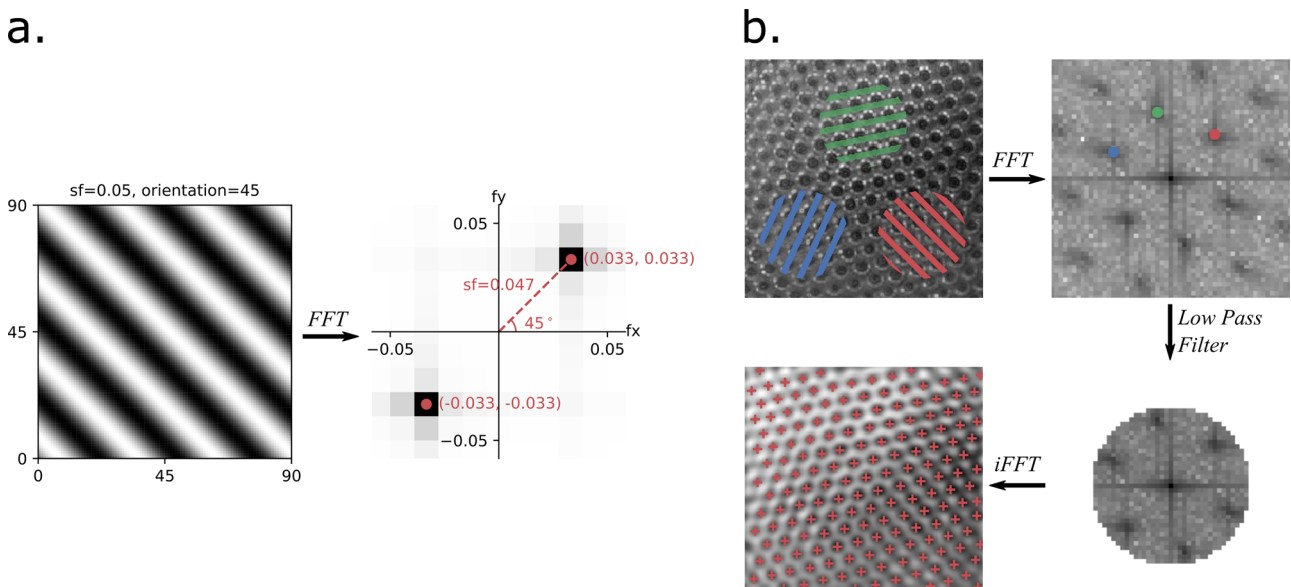

**Fig. 4 The ommatidia detecting algorithm (ODA) extracts periodic signals in a 2D image using the FFT. a** A 2D sinusoidal grating with a spatial frequency of .05 and orientation of 45° (left) and its reciprocal image (right). In the frequency domain of a 2D FFT, called the reciprocal space, gratings are represented by an x- and y-frequency. The polar coordinates represent visual properties of the corresponding grating. The radial distance is a grating's spatial frequency, with high frequencies farther from the origin. The polar angle is the grating's orientation, which is perpendicular to the grating's wavefront. Notice that the reciprocal space has local maxima (in red) approximately equal to the input grating parameters (polar angle=45° and radial distance = .047 ± .005). **b** The ODA pipeline for finding ommatidial centers. In a hexagonal lattice, there are three major axes (here in blue, green, and red). Each axis corresponds to a 2D spatial frequency (and it's negative), visible in the image's reciprocal space. The periodic nature of the axes results in harmonic frequencies. A low-pass filter returns a version of our original image primarily representing these three axes. The center of each ommatidium is found at the local maxima of the filtered image.

We obtained micro-computed tomographs (μCTs) of a fruit fly (*D. mauritania*), tobacco hornworm (*Manduca sexta*), elephant hawkmoth (*Deilephila elpenor*), and honeybee (*Apis mellifera*). The fruit fly μCT was collected by Maike Kittelmann and used with her permission. The head was fixed and dehydrated among other heads in the same way as SEM samples. Once in 100% ethanol, they were stained with 1% Iodine in ethanol before scanning at the TOMCAT beamline of the Swiss Light Source (Paul Scherrer Institute, Switzerland). Fly heads were placed into 10 μl pipette tips in 100% Ethanol and scanned using a 16 keV monochromatic beam with a 20 μm LuAG:Ce scintillator. For more information on the scanning procedure, see Torres-Oliva et al. (2021)[43]. Vouchered moth specimens from the Florida Natural History Museum were stored at -20°C in 95% ethanol, then heads were sliced, with antennae removed, and soaked in staining solution (I$_2$ + KI, equal proportions 1.25% I$_2$ and 2.5% KI solutions) in Eppendorf vials or falcon tubes for 36–48 h. *M. sexta* was scanned with a Phoenix V | Tome|X M system with: a 180kv x-ray tube, a diamond-tungsten target, 80 kV tube voltage, 110 μA current, 17.8 mm source object distance, 793 mm object-detector distance, and capture time adjusted to maximize absorption range for each scan. The acquisition consisted of 2300 projections, 8 s each. GE's datos|x r software version 2.3 processed raw x-ray data, producing voxel size of 4.50074 μm. Volume files were imported into VG StudioMax version 3.3.3 (Volume Graphics, Heidelberg, Germany), eyes isolated with the segmentation tools, then exported as Tiff stacks. *D. elpenor* was scanned with a Zeiss Xradia 520 Versa (Carl Zeiss Microscopy GmbH, Jena, Germany), with: 80 kV tube voltage, 88 μA current, low energy filtering, 22.5 mm source object distance, 210 mm object-detector distance, an indirect detector comprising a scintillator, a 0.392x optical lens, and a camera provided to us by Deborah Glass. The acquisition consisted of 3201 projections, 8 s each, with the adaptive motion correction option in Scout-and-Scan software (Carl Zeiss Microscopy GmbH). The tomographic reconstruction automatically generated a 32-bit txrm set of tomograms with an isotropic voxel size of 3.3250 μm. The XRM controller software (Carl Zeiss Microscopy GmbH) converted data to a stack of 16-bit tiff file. A previously scanned honeybee eye[28,37] was downloaded from *Morphosource* at: https://www.morphosource.org/concern/parent/000396179/media/000396182. Voxel size: 5.0 μm. This sample was scanned with a synchrotron x-ray source[31]. See[37] for the details.

**Data validation and manual counts**. For the microscope images, ommatidial counts and diameters were measured manually by the researchers providing the datasets (Ravi Palavalli-Nettimi measured the ant eye replicas, Maike Kittelmann the SEMs, and John P. Currea the *D. melanogaster* micrographs). Image processing software like ImageJ allowed us to manually annotate each ommatidium and estimate their diameter as the average of several diameters taken across the eye. Our approach was different for the various CT datasets. For the *D. mauritania* and *A. mellifera* datasets, we compared our results to previous measurements taken on the same scan. Note that for *D. mauritania* this was a direct count but was a density-based estimate for *A. mellifera*. Density-based estimates approximate the ommatidial density across the surface of the eye by taking local measurements and averaging across the whole eye surface. Our other CT stacks, *D. elpenor* and *M. sexta*, are new and don't have direct measurements in the literature, so we compared our results to density-based estimates of other conspecifics.

**Ommatidia detecting algorithm**. The fourier transform is a mathematical transformation that decomposes arbitrary functions into component sinusoids, which can highlight periodic or repeating patterns in a signal. For digital images, the sinusoidal elements of a 2D Fast Fourier Transform (FFT) are plane waves (gratings), characterized by contrast, frequency, phase, and orientation (Fig. 4a). Operations applied to the frequency representation (reciprocal space) can be inverse-transformed to generate a filtered image. The hexagonal arrangement of typical ommatidia has 3 major axes (Fig. 4b), each approximated by a grating, and filtering frequencies higher than these generates a smooth image, with maxima near ommatidia centers. The inverse of these frequencies, approximating the ommatidial diameter, also provides useful bounds for easily applying local maxima detection algorithms to the smoothed image. In particular, our program searches for maxima within 25% of the FFT-derived ommatidial diameter, which we found to be robust even for less regular ommatidial lattices.

We developed a Python language module, the ommatidia detecting algorithm (ODA), which: (1) generates a 2D FFT, (2) finds the three fundamental gratings as the local maxima closest to the reciprocal image center, using autocorrelation to amplify periodic elements, (3) filters higher image frequencies, (4) inverts the filtered 2D FFT, and (5) finds local maxima in the smoothed image (Fig. 4b). There are several options when running this that are described in depth in the documentation. Importantly, the ODA can check for just the first 2 instead of the 3 fundamental frequencies, in principle allowing the program to work on ommatidia arranged in a square lattice such as that found in the reflecting superposition eyes of decapod crustaceans[62]. This option also helps for noisy images where the highest fundamental frequency is sometimes mistaken as a harmonic of one of the other two. For instance, although we used the default settings for all other results, we used this option on the dataset of 29 *D. melanogaster* micrographs, resulting in much more accurate results than those found without selecting this option. Also, users can check the results in the reciprocal image with maxima superimposed using a graphical user interface we developed (Supplementary Fig. 1A). The

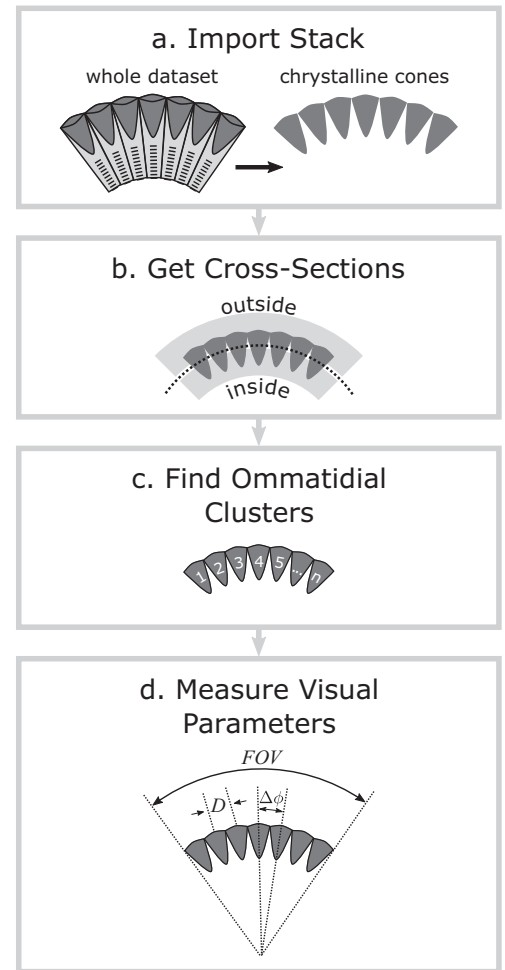

**Fig. 5 General steps along the ODA-3D pipeline. a** ODA-3D starts with an image stack, which may be pre-filtered and cleaned of unrelated structures. **b** Then we filter the relevant density values and fit a surface to the coordinates, allowing us to split the points into two sets or cross sections, inside and outside the fitted surface. **c** For each cross section, we generate spherically projected, rasterized images that are processed by the FFT method for locating ommatidia, yielding approximate centers for the ommatidia. With these centers, we can find the coordinate clusters corresponding to independent ommatidia **d** These can then be used to automatically measure eye parameters.

program stores ommatidia coordinates and calculates ommatidial diameter. An optional imported mask (a white silhouette on a black background) can help avoid false positives outside of the eye.

**Measuring ommatidia using μCT**. We have further used the ODA to process 3D μCT data. This pipeline: (1) imports a stack of images representing a 3D dataset of points in the crystalline cones (Fig. 5a) or corneal lenses. Images can be edited to delete irrelevant pixels, and our program can prefilter data by choosing a density range in a graphical interface, to further isolate crystalline cones, and can preview the whole dataset (Fig. 6A). Supplementary Fig. 1A shows a frame of the 3D user interface offered by our program.

(2) projects coordinates onto 2D images processed by the ODA (Fig. 5b). The layer of crystalline cones curves with little variation normal to its surface, allowing a sphere fit with least squares regression. The algorithm transforms points to spherical coordinates and interpolates a continuous surface, modeling the points' radii as a function of elevation and azimuth. Finally, it selects the cross-sectional surface containing 50% of residuals (Fig. 6b). Supplementary Fig. 1B shows the residual distance of each point from this surface, which is optionally displayed at the end of this stage of the program. This allows you to check that the surface fit is not biased to any particular region of the eye.

(3) forms images of this cross-sectional sheet by taking 2D histograms of elevation and azimuth (as in Fig. 5c but ignoring residual distance) in 90°X90° segments (although other window sizes can be specified). Processing the surface in

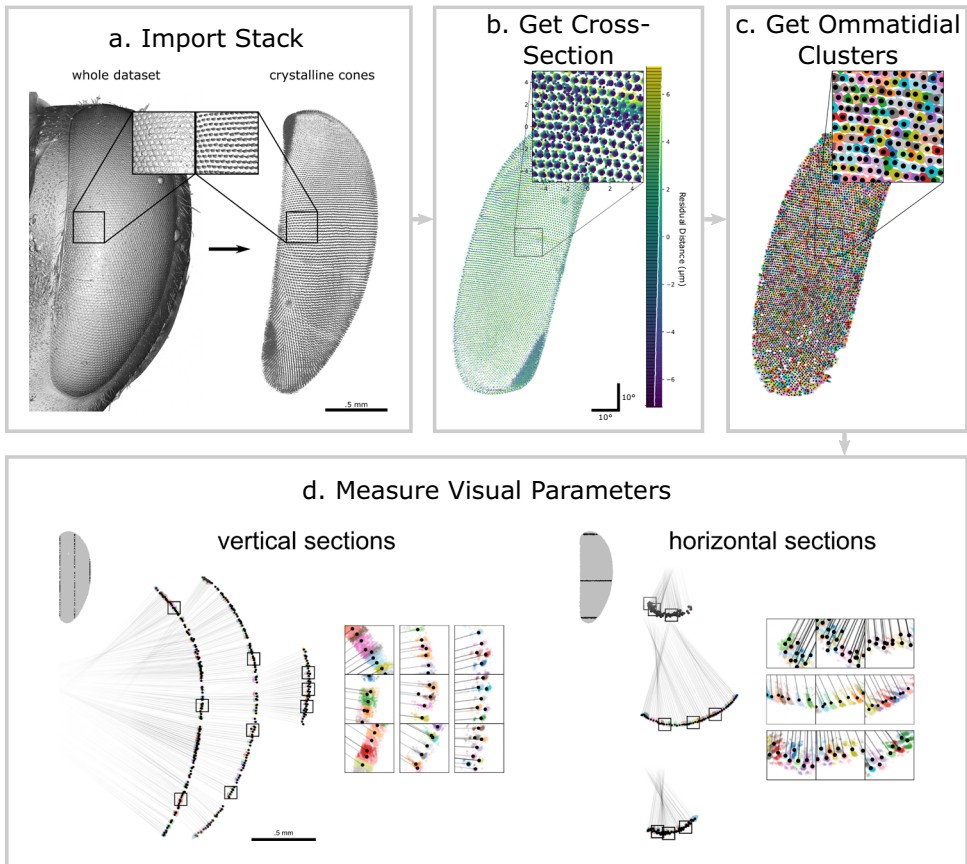

**Fig. 6 Checkpoints along ODA-3D for the honeybee scan. a** *Import Stack*: The pipeline started with an image stack that we pre-filtered and manually cleaned of unrelated structures. **b** *Get Cross-Section*: A surface was fitted to the coordinates identifying a cross-section of points within 50% of the residuals. We generated a spherically projected, 2D histogram of the crystalline cone coordinates colored by their residual distance from the cross-section. **c** *Find Ommatidial Clusters*: We apply the ODA to label individual crystalline cones. The ODA locates the cluster centers, allowing us to partition the coordinates based on their distance to those centers (boundaries indicated by color-filled polygons). Then we cluster points based on their nearest center and apply our custom clustering algorithm to improve segmentation of more skewed ommatidia. **d** *Measure Visual Parameters*: Ommatidial diameter corresponds to the average distance between clusters and their adjacent neighbors. Ommatidial count corresponds to the number of clusters. For non-spherical eyes, we can partition the 3D coordinates into vertical (*left*) and horizontal (*right*) cross-sections in order to calculate independent vertical and horizontal anatomical IO angle components accounting for skewness in the ommatidial axes. The insets zoom into regions around the 25th, 50th, and 75th percentile ommatidia along the y-axis (*left*) and x-axis (*right*). Note: in **c** and **d** the different colors signify different crystalline cone clusters. In **d**, the black lines follow the ommatidial axes and the black dots indicate the cluster centroids. Also in **d**, note how some ommatidia in the periphery (for example, horizontal insets row 2, columns 2 and 3) were erroneously segmented and thus underestimate their skewness and IO angles.

smaller segments and recentering before forming each image avoids extreme spherical warping and improves the accuracy of the algorithm. The ODA approximates lens centers within each segment, then recombines them into the original coordinate system. Finally, each point is labeled by its nearest center, effectively segmenting points into clusters corresponding to separate ommatidia (Fig. 6c). Although this works well for spherical eyes, when ommatidia are substantially skewed the overlap of projected clusters prevents the algorithm from segmenting correctly, often splitting up proper clusters among their neighbors. Optionally, our program can identify these problematic clusters and attempt to correct them using a custom clustering algorithm. Problematic clusters are identified as those with a large proportion (≥10%) of elements whose nearest neighbors were elements of a different cluster. For each problematic cluster, the algorithm finds the immediate neighborhood of 6 clusters and applies a nonlinear minimization algorithm (Scipy's differential evolution) to minimize both 1) the mean projected distance between each point and its cluster center and 2) the aspect ratio of the cluster's longest side to its shortest in 3D. Because the clusters are approximately parallel on a small scale and elongated along the ommatidial axis, this process often converges on the plane orientation orthogonal to the clusters' longitudinal axes. The program only incorporates these cluster corrections if they result in substantially less problematic elements (<5%). This dramatically improves cluster segmentation for nonspherical eyes like the honeybee scan and adds a trivial amount of time to the program's total duration. Optionally, our program also allows the user to manually edit cluster centers (Supplementary Fig. 1C–D) and then displays the outcome of this segmentation (Supplementary Fig. 1E)

(4) measures visual parameters of the eye (Fig. 5d). The distance between adjacent centroids approximates lens diameter. The ideal ommatidial axis is derived from planes formed by triplets of centers near the cluster, and we approximate the surface normal by averaging the normal vector for each plane. Singular value decomposition finds the semi-axes of the ellipsoid of a cluster and our program selects the one closest to the ideal axis to estimate the anatomical ommatidial axis. The angular difference between ideal and anatomical axes estimates anatomical skewness.

Anatomical axes of neighboring ommatidia should yield reliable IO angles, but the raw axes are highly variable. While greater resolution, or using other structures to extend approximations closer to the intersection, could improve accuracy, our program reduces variability by replacing each anatomical axis with the average axis of a local neighborhood of clusters (akin to the rosette averaging procedure used in[54]). The program allows the user to determine the radius of this neighborhood in terms of ommatidial diameter: a radius of 1 specifies immediate neighbors within one diameter, 2 specifies immediate neighbors and immediate neighbors of those, and so on. Here we used a radius of 5. In general, this has the effect of reducing local variability in IO angles while maintaining global patterns. However, it also skews axes close to the boundary of the eye towards the center due to the asymmetrical participation of neighboring points, thus reducing FOV estimates. As a compromise, in addition to allowing the user to determine the neighborhood radius, our program uses the neighborhood-averaged angles for IO angle measurements but the raw axes for measuring the projected FOV.

To calculate anatomical IO angles, our program partitions coordinates into evenly spaced vertical and horizontal sections (Fig. 6d), in which we projected

clusters onto a parallel plane. For instance, for vertical sections, all clusters within a range of x values are considered and the 2D clusters formed by y and z values determine the vertical angle component. Angles are then calculated along the 2D plane using the locally averaged axes described above. This repeats independently for each vertical and horizontal section. The process approximates a horizontal and vertical subtended angle for each cluster pair and calculates the total angle as their hypotenuse. We call this hypotenuse the anatomical IO angle. This method allows independent approximations for horizontal and vertical IO angles across the eye, and by keeping track of cluster pair orientation along the eye surface, we can calculate horizontal and vertical IO angles using the two-dimensional lattice proposed by[30] (illustrated in Fig. 3c).

Finally, the program generates two spreadsheets: (1) for each crystalline cone cluster, the Cartesian and spherical coordinates of the centroid, the number and location of the points within its cluster, the approximate lens diameter, and the ideal and anatomical axis direction vectors are saved per row; and (2) for each pair of adjacent crystalline cones, the cones' indexes from spreadsheet 1, the center coordinates for the two cones, the resulting orientation in polar coordinates, and the anatomical IO angle between the two are saved per row. These allow approximations of how spatial acuity, optical sensitivity, and the eye parameter[60] vary across the eye[37]. The full code for measuring ommatidia with μCT is available at GitHub where you can download the Python package and basic examples on how to use it (see Data Availability section).

**ODA testing**. To validate performance and speed of the ODA software, we applied the ODA to each image after programmatically lowering image resolution (by bin-averaging the images into larger and larger bins) and contrast in order to determine the performance constraints of the ODA (by reducing the 8-bit range of brightness values to narrower and narrower distributions). We recorded the output omma-tidial count and diameter for each image and deterioration level and compared them to manual estimates of the same. Using manual measurements of ommatidial diameter, we converted the image resolution into units of pixels per diameter. Runtime was measured using the cProfile module in Python and the degradation experiments were run on a Microsoft Surface Pro laptop with an 11th Gen Intel(R) Core(TM) i7-1185G7 @ 3.00 GHz processor and 15.8 GB of usable RAM.

**Statistics and reproducibility**. For the most part, we report descriptive statistics to show the accuracy of our program. As in Table 1, variables that follow a generally symmetrical distribution are reported as a mean ± the standard deviation and those with skewed distributions, like IO angle, are reported as the median (interquartile range). When reporting correlations, such as between manual and automatic measurements, we report the Pearson correlation and the corresponding p-value of a two-tailed F-test.

**Reporting summary**. Further information on research design is available in the Nature Portfolio Reporting Summary linked to this article.

## Code availability

The ODA module, written in Python, is available for download: www.github.com/jpcurrea/ODA/. The version of the package used in this paper is available in our online dataset at https://doi.org/10.6084/m9.figshare.21521142[61].

## Data availability

All datasets and code are freely available at https://doi.org/10.6084/m9.figshare.21521142[61]. The image stack of the μCT data for the *A. mellifera* scan was drawn from Taylor et al. (2018)[37] and is available at https://www.morphosource.org/Detail/ProjectDetail/Show/project_id/646[62].

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

## Acknowledgements
This is contribution #1519 from the Institute of Environment at Florida International University. This research was supported by grants from the National Science Foundation: IOS-1750833 to JT, BCS-1525371 to JC, DEB-1557007 to AYK and JT and IOS-1920895 to AYK. The content is solely the responsibility of the authors and does not necessarily represent the official views of the National Institutes of Health or the National Science Foundation. We thank members of the Kawahara Lab at the Florida Museum of Natural History, McGuire Center for Lepidoptera and Biodiversity, for raising and vouchering moth specimens. We thank the University of Florida CT facility for helping acquire the *M. sexta* CT scan, and Kelly Dexter for help with sample staining. We thank Deborah Glass and Ian Kitching from the Natural History Museum of London for helping with the CT procedure and sharing the *D. elpenor* CT scan, which study was funded by NERC grant number NE/P003915/1 to IJK. We thank Maike Kittelmann for sharing her SEMs and µCT of two fruit fly species (*D. melanogaster* and *D. mauritiana*) and for advice in using available 3D software. Gavin Taylor and Emily Baird shared their CT scans of *A. mellifera*. Michael Reisser and Arthur Zhao provided valuable discussions regarding the µCT pipeline.

## Author contributions
J.C.: Conceptualization, Methodology, Software, Visualization, Writing- Original Draft. Y.S.: Conceptualization, Methodology, Writing- Review and Editing, Data Collection and Curation. J.T.: Conceptualization, Methodology, Writing- Review and Editing, Supervision, Funding acquisition. A.K.: Conceptualization, Methodology, Writing- Review and Editing, Funding acquisition.

## Competing interests
The authors declare no competing interests.
