## [Peer Review File · Communications Biology]

Reviewers' comments:

Reviewer #1 (Remarks to the Author):

The work presented in the manuscript COMMSBIO-21-0309-T offers a software solution to the technical problem of counting and measuring ommatidia in the invertebrate compound eye, and evaluating structural variables related to visual acuity, sensitivity and field of view. The authors use standard microscopy and ex vivo micro computed tomography digital images of compound eyes of several insects.

The paper is correctly written. The software main virtues are well described, but the algorithmic details cannot be evaluated properly because the programs and samples are not accessible to the reviewers. In any case, I feel it is not easy for an average biologist to follow the explanation of the digital processing performed with the ODA module to analyze 2D and 3D ommatidia images, and Figs. 3 and 4 are not very helpful either. The authors should try to be more didactic for their work to appear in a multidisciplinary journal like Comm. Biol.

A major concern is the lack of experimental comparison with other potentially competing automated or semiautomatic analytic algorithms. The authors mention several published techniques that analyze digital images of invertebrate ommatidia, but discard them without a proper comparison because "they were not validated against manual measurements", a process that I would not consider an exact counting when coming from out-of-focus eye boundary images. The authors actually find "overestimations" of ommatidia in their own ODA counts! Moreover, the authors also censure previous ommatidia analysis techniques focused on *Drosophila* eyes because "they were not tested on different invertebrate species". They might test them with their own images.

I think the paper would benefit from an experimental comparative approach with existing methodologies to persuade researchers in this field that ODA is of special value for their experiments. In this sense, several other approaches have been published that authors fail to cite, compare and discuss. Some of them are incorporating machine and deep learning techniques to fully automatize the analysis (doi:10.3389/fnins.2020.00516). Others (doi:10.1242/jeb.210195) even deal with micro-CT scans of arthropod eyes to study their visual spatial resolution.

Several other minor issues can be considered to improve their work:

- It would be useful to address the level of irregularity of the cornea array that might invalidate a ODA analysis.
- Researchers should know the resolution / bit-depth range of 2D images well resolved with ODA. I found no mention of the features of images used for ODA counts and measures. The authors confirm in the Discussion section that "images for the ODA require sufficient resolution" and give an estimate of half the smallest ommatidial diameter, but they should assess the error involved in counting and measuring ommatidia based on different resolutions and sample sizes.
- Also, it might be useful to check the algorithm with scanning EM digital images, a microscopy technique widely used when studying compound eyes.
- Although ODA-3D appears to work fine with microCT digital images, this technology is not readily available for every lab working on arthropod compound eyes. It would be nice to know whether (and how accurately) ODA-3D works with standard histological sections and Z-scan confocal images.

Reviewer #3 (Remarks to the Author):

The authors tackle the time-consuming and tedious job of counting and measuring ommatidia in arthropod eyes. They present two novel open source Python algorithms for automatically quantifying externally and some internally measurable or calculable parameters of the compound eye using 2d and 3d datasets, the latter which is increasingly being generated for anatomical study. They test their algorithms (ODA and ODA-3D) using light micrographs and CT-scanned 3d data sets from previous studies, comparing the results to their own manual measurements and counts. A few insect species were used, varying in size and eye shape and design. They also compare their results to equivalent values in the literature and critically consider the limitations of the automated algorithms.

The algorithms provide a welcomed automation of a tedious data collection process for many compound eye researchers. However, it is not completely free from errors and difficulties associated with the data image quality. Using example data from ant and fruit fly eyes, they show that their ODA gives similar, but not perfectly matching results to manual ommatidia counts and diameter measurements from 2D micrographs, provided the eye is more or less flat. Difficulties would arise when making these measurements from a single image of an eye with very curved profile. The second program, the ODA-3D, appears to work reliably in the spherical eyes of two moth species, and to some extent successfully in the vertically elongated oval shaped eyes of a bee, but it was not clear how much manual tweaking and time the program needs for the user to adjust to different, more challenging data. There were issues with varying scan contrast for the bumble bee whereby crystalline cones were unsuccessfully segmented in the dorsal eye, resulting in missing data. This problem may be commonplace with other large data sets and is a difficult one to navigate without manual tweaking. The program appears to work well for good quality scans of eyes with spherical shape. They mention that for large asymmetric eyes, there are still some time consuming stages (e.g. spectral clustering) of the algorithm. Much compound eye research concerns animals with interesting optical specialisations, such as acute zones (briefly mentioned). It would be interesting to know how well the ODAs perform with these in future.

Ideally, one would have liked to see a few more examples of the program in practise, although I appreciate that there is not a wealth of 3d data sets available currently. It would be interesting to know if the authors can you speculate (or better still, test) whether it work as effectively with animals like crustaceans? These animals tend to possess flat surface corneal lenses, where outward patterns and features (edges of ommatidia) may be less defined in the images? Decapods often have eyes that wrap all the way round an eyestalk – can you speculate on how the ODA-3D will cope with that? I also look forward to seeing if the programs (in future) can be adapted to work with eyes that have irregular ommatidia, lacking a hexagonal lattice distribution, or animals such as shrimp with square facets.

There will be undoubtedly some small errors in the calculations, accumulating from the steps involved, but the programs appear to work well at least for quick* approximations and comparisons between species etc., (*provided the user is able to use Python). The error in measuring the thousands of very small IO angles in non-spherical eyes was particularly apparent in the oval eyes of *B. terrestris*, producing very inaccurate results. The authors have identified this and come up with a solution, to separate them into independent horizontal and vertical components and sample pairs within 15 degree centroids. This reveals interesting IO differences in azimuth / elevation across the eye that seem to have been missed by others previously.

Specific comments:

1. General = stating that “compound eye being the most common/ prevalent eye design” is repeated many times throughout the manuscript. In the Introduction, the rationale for this study is a little unsatisfying – yes compound eyes are very prevalent, but can you say a little about why is it important to understand them and obtain the data that this work facilitates? Perhaps add a couple of examples of arthropod eye research leading to useful innovations e.g. camera design, pest control traps etc.

2. Introduction: Another paper to be aware of and reference, which presents a Matlab programme to achieve similar goals: Bagheri et al (2020) <https://jeb.biologists.org/content/223/1/jeb210195>
3. Line 31 = "light level" - change to light intensity and spectral characteristics
4. Line 34 = "individual facets, (called ommatidia)" Not strictly true, the facet lenses are the external and outwardly visible part, which contribute to each ommatidium
5. Line 39 = "many structures" - needs slight rewording, as the only structures that are externally visible are corneal lenses and (sometimes) pseudopupil, which do reveal some insight into optical capabilities in a way camera eyes do not.
6. Line 41 = worth noting that the light reception inside each ommatidial field of view is averaged and does not produce a tiny image.
7. Line 55 = "Ommatidia count and diameter can be derived from 2D images" This only works in instances where the eye curvature is slight so that all facets are angled toward the camera/detection chip. There must be difficulties in counting and measuring facets at the eye edge by any means.
8. Lines 77 & 579 = not all superposition eyes are spherical. Some krill and moths present exceptions with asymmetrical shapes. See:
Land, M.F., Burton, F.A. & Meyer-Rochow, V.B. (1979). *J. Comp. Physiol.* 130, 49.
Warrant E, Bartsch K, G Nther C. (1999). *J Exp Biol.* 202 (Pt 5):497-511.
9. Line 80 = skewed ommatidia, can you clarify a bit more about this or provide an example / reference. It is not clear if you mean the corneal surface lies at a skewed angle from the crystalline cone and/or photoreceptors. It would be beneficial (for those unfamiliar with this phenomena) if you could briefly describe what happens to light entering these facets and how the aperture is effectively reduced, resulting in lower sensitivity.
10. Line 105 = Specimens and eye imaging
11. Line 106 = what material were the eye molds made from?
12. Lines 114-5 = "I2" subscript needed
13. Line 121 = Are these many decimal places accurate / necessary for voxel size?
14. Line 132 = voxel size would be worth stating here for the synchrotron data as a comparison to microCT.
15. Methods = Brief description to say how you made your manual measurements would be good, especially if any software were used to help the process.
16. Fig. 3 panels A-D not referenced in legend.
17. Line 146 = how are ommatidial diameters calculated and what happens when a facet is missed - e.g. Fig 5a, 4th panel?
18. Lines 149-153 = How reliable and time consuming is this step for large or complex eyes, or scans with noise/ poor contrast? Are crystalline cones always contrasting enough in intensity to segment from other eye structures easily across different species and data acquisition modes? Often crystalline cones, being soft and gelatinous in nature, suffer deformations in preserved eyes. Many eyes have accessory pigment cells close by between the CCs that may have similar intensity values in the tomograms. I see you had issues with the dorsal region of the bumblebee eye (line 240), resulting in missing data. Is there any way around this - say to match the facet centres with their underlying photoreceptor column instead?
19. Fig. 4 = scale bar would be better in microns
20. Line 209-13 = The program appears to create a good approximation of facet lens count and diameter from 2d images, but is not completely reliable/ accurate. Is this due to image contrast differences? Do shadows and reflections pose a problem for recognising the facets?
21. In some cases where microscope images of intact eyes are used, miscalculations might also arise because of how the 2d images represent the ommatidia at the borders of the eye, due to their side-on angle with respect to the camera. Manual counts would also be erroneous for the same reason. Fig. 5c shows that lens centres appear to cluster together at the edges due to the eye curve. Is this affecting your facet diameter calculations?
22. Fig. 5A. Facets missed by the program appear obvious to the human eye due to pattern recognition, could this be incorporated into the algorithm somehow to prevent under-sampling?
23. Fig. 5B. It would help to have an x=y line or grid background across the figures to facilitate visualisation of how many points lie above / below the lines for comparison.

24. Line 226 = "Facets actually curve slightly with the curvature of the eye" Yes, for light micrograph images of intact eyes, only facet diameters from the central region of the eye facing toward the camera can be accurately measured. How were manual measurements made? Lenses tend to become smaller at the edges of an eye, where their profile will be side-on and difficult to measure in a single 2d image by any means.

25. 235 = "The 2D histograms and the superimposed ommatidial centers suggested that the first count was more accurate for *M. sexta* while the second count was more accurate for *D. elpenor*." - I was slightly confused here by your comparisons of two sets of counts, as it was not mentioned previously that you ran the program on the same data twice, if that is what you meant?

26. Lines 242-250: Comparisons with literature values. Ommatidia counts will vary naturally across individuals of a species, so this isn't a strong evidence for accuracy. It doesn't compare direct measurements so could perhaps be more of a discussion point rather than presented in the results.

27. Lines 268- "However, instead of using the center of the head as our center, we used the center found in step B of ODA-3D, which is near but not exactly the center of the head."

28. Why was an off-center point used and does this apply to all eyes processed this way? Does it perhaps approximate to the centre of the basal membrane or something? Also, the sentence is a bit clunky to read with the word "center" used 4 times.

29. Lines 297 and 302 = Figure B?

30. Line 299 = These comparisons to measurements by Baumgärtner (1929) seem a little bit tenuous, due to lack of species information. Are there any (recent!) sources of data with known bee species you can compare to?

31. Line 302 = reword slightly so that is clear from the start that the "previous measurements" refer to values published in another study (+give ref). In general it might be worth replacing "previous" with the name of the authors to avoid confusing with your own previous measurements.

32. Fig. 6 & 7 = include animal species in the legends

33. Fig. 7b = in the table, top row, are the two count values separated by – symbol a range or two separate values that you calculated from running the programs twice? The range in values seems very high, many hundreds of ommatidia different.

34. Line 332 = End of Results could perhaps be reorganised "Similar tradeoffs" – perhaps this is referring to the skewness rather than fixation and binocular FOV just stated. Worth adjusting for readability.

35. Line 364 = "As far as we know, this has not been measured previously, though it is consistent with previous measurements..." could do with re-wording!

36. Line 392 = Add in that there may also be glial pigment cell interactions between the crystalline cones or deformation of the soft cells, which may present difficulties in auto segmentation in this region of the eye.

Referee expertise:

Referee #1: eye classifier, ommatidia, plugins

Referee #2: microCT, eye

Reviewers' comments:

Reviewer #1 (Remarks to the Author):

The work presented in the manuscript COMMSBIO-21-0309-T offers a software solution to the technical problem of counting and measuring ommatidia in the invertebrate compound eye, and evaluating structural variables related to visual acuity, sensitivity and field of view. The authors use standard microscopy and ex vivo micro computed tomography digital images of compound eyes of several insects.

Comment: The paper is correctly written. The software main virtues are well described, but the algorithmic details cannot be evaluated properly because the programs and samples are not accessible to the reviewers. In any case, I feel it is not easy for an average biologist to follow the explanation of the digital processing performed with the ODA module to analyze 2D and 3D ommatidia images, and Figs. 3 and 4 are not very helpful either. The authors should try to be more didactic for their work to appear in a multidisciplinary journal like Comm. Biol.

Response: *We revised figure 3 and 4 to explain the process visually in a manner that should enable non-specialists to understand it better. We have also revised the text under the methods section “Measuring Ommatidia using μ CT”. The code has been revised significantly and the program is faster, more streamlined and now includes features such as data visualization and the ability to resume the analysis at different steps. Moreover, the program has been successfully tested by several researchers from other labs. All code and instructions are also available publicly in github.*

Comment: A major concern is the lack of experimental comparison with other potentially competing automated or semiautomatic analytic algorithms. The authors mention several published techniques that analyze digital images of invertebrate ommatidia, but discard them without a proper comparison because “they were not validated against manual measurements”, a process that I would not consider an exact counting when coming from out-of-focus eye boundary images. The authors actually find “overestimations” of ommatidia in their own ODA counts! Moreover, the authors also censure previous ommatidia analysis techniques focused on *Drosophila* eyes because “they were not tested on different invertebrate species”. They might test them with their own images.

I think the paper would benefit from an experimental comparative approach with existing methodologies to persuade researchers in this field that ODA is of special value for their experiments. In this sense, several other approaches have been published that authors fail to cite, compare and discuss. Some of them are incorporating machine and deep

learning techniques to fully automatize the analysis (doi10.3389/fnins.2020.00516). Others (doi:10.1242/jeb.210195) even deal with micro-CT scans of arthropod eyes to study their visual spatial resolution.

Response: *The reviewer raises a fair point about comparing our work to previous methods—we had attempted this prior to submitting—however many of the published methods had issues that prevented replicating their results or testing their methods with existing data, in some cases the code was not even public. This is often the case with older methods but we still elaborate in more detail the attempts we made to compare our tool to other similar methods.*

1) *We read doi10.3389/fnins.2020.00516 when working on the manuscript and tried to use it for our project. Upon close inspection, we found that the program is not designed to count ommatidia or measure their diameters, and would require further work to do so. Instead, as stated in the abstract, it “provides a robust quantification framework that can be generalized to address the loss of regularity in biological patterns similar to the Drosophila eye surface and speeds up the processing of large sample batches.” This process involves choosing a region of interest (not the whole eye) in order to measure the regularity of the ommatidial lattice and does not directly provide ommatidial count or diameters. Further, their code was not freely available and their machine learning algorithm had been trained and tested only on images of Drosophila melanogaster. Lastly, we have messaged the authors and found that additional work would be needed to generate ommatidial counts and diameters.*

2) *Similarly, the code and software from doi:10.1242/jeb.210195 were not made public, so we messaged the authors directly. They confirmed that the code is not publicly available and mentioned that running it would require some training. Moreover, the algorithm was applied only to scans of the eyes of fiddler crabs.*

As a solution, in the introduction, when we compare the available alternatives, we state:

“Although several algorithms and software plugins have been proposed to extract ommatidia, they require user input for each image, and underestimate ommatidia counts (Woodman, Todd, and Staveley 2011), overestimate ommatidial diameter (Schramm et al. 2015), or were not validated against manual measurements or measurements in the literature (Diez-Hernando et al. 2015; Iyer et al. 2016; Vanhoutte, Michielsen, and Stavenga 2003). In addition, none are tested on multiple species, over a substantial range of eye sizes, or with different media.”

Comment: *The authors actually find “overestimations” of ommatidia in their own ODA counts! Moreover, the authors also censure previous ommatidia analysis techniques focused on Drosophila eyes because “they were not tested on different invertebrate species”. They might test them with their own images.*

Response: *To clarify, we did not discard Woodman et al. (2011) for just underestimating counts or Schramm et al. (2015) for just overestimating diameters. These are valid criticisms of our own method, which we presented openly in the manuscript. We discarded the two programs because in addition to these errors they require human*

input on an image-by-image basis. Admittedly, our algorithm is not entirely automated since it often requires generating a masking image to indicate the pixels to include in the input. However, their programs require user input for various stages of the actual algorithm per image. In addition, the programs were not publicly available when we tried to use them. This was true of all of the articles listed there. In fact, Schramm et al. (2015) doesn't propose a program of its own at all, just a series of instructions to use ImageJ for this purpose. In conclusion, our objections to these alternatives still stand.

To ensure our tool does not suffer similar issues, we went through several rounds of beta testing and have added unit testing and sample data that users can use to test if the method works. Moreover, our program offers a graphical user interface to manually add or delete ommatidial centers superimposed on the original image to account for any of these errors.

Minor comments

Comment: - It would be useful to address the level of irregularity of the cornea array that might invalidate an ODA analysis.

Response: *We include the following text in the discussion*

“Because the ODA depends on spatial frequencies corresponding inversely to the ommatidial diameter, an eye with a wide range of diameters may not work, and the ODA should be tested on eyes containing acute or sensitive zones, such as the robberfly (Wardill et al. 2017) and lattices with transitioning arrangements, such as hexagon to square in houseflies (Stavenga 1975) and male blowflies (Smith et al. 2015). Likewise, the ODA-3D should be tested on non-spherical non-oval eyes. While our program appropriately measured anatomical IO angles across the honeybee's oval eye and actually corroborated its oval eye properties, it may not work when IO angles change dramatically like in the robberfly (Wardill et al. 2017). Finally, the ODA-3D should be tested on non-insect arthropods.”

We also added a feature in the software to allow the user to try two options for finding the peak frequency assuming either a regular or irregular lattice.

Comment: Researchers should know the resolution / bit-depth range of 2D images well resolved with ODA. I found no mention of the features of images used for ODA counts and measures. The authors confirm in the Discussion section that “images for the ODA require sufficient resolution” and give an estimate of half the smallest ommatidial diameter, but they should assess the error involved in counting and measuring ommatidia based on different resolutions and sample sizes.

Response: *This is an excellent point. We've addressed this by adding a substantial amount of work to the results. We applied the algorithm to the following sets of images: SEM images of D. melanogaster and D. mauritania eyes taken by Maike Kittleman, and the ant eye replicas and direct micrographs of D. melanogaster eyes already discussed in the text. We measured the success of the ODA as a function of resolution and contrast. The resolution limit is defined by the Nyquist limit, requiring at least 2 pixels*

per ommatidium and contrast has a small effect that's only clear after 10- to 100-fold decreases in contrast.

Comment: Also, it might be useful to check the algorithm with scanning EM digital images, a microscopy technique widely used when studying compound eyes.

Response: *We agree with the point raised and as mentioned above, we've now added SEM images of two Drosophila species showing that while the program does work generally, it tends to miss ommatidia near the boundary due to the curvature of the eye, like we found with the micrographs.*

Comment: Although ODA-3D appears to work fine with microCT digital images, this technology is not readily available for every lab working on arthropod compound eyes. It would be nice to know whether (and how accurately) ODA-3D works with standard histological sections and Z-scan confocal images.

Response: *In principle, the algorithm should work on those media as well, as long as the image resolution is sufficiently high. The resolution is theoretically limited by the Nyquist limit set by the spacing of the smallest ommatidia, requiring at least 2 pixels per ommatidial diameter. In practice, we found that this limit didn't depend on the imaging medium, is a little over 4 pixels per ommatidium, and is surprisingly robust to changes in contrast, often working on images where we can't even see the ommatidia. We have now shown that it works on direct micrographs of eyes, micrographs of eye replicas, SEM images of eyes, and CT scans of eyes. Unless there are theoretical reasons to doubt its application to histological and confocal images, we believe such tests of the ODA would be better suited for future versions of the program.*

Reviewer #3 (Remarks to the Author):

The authors tackle the time-consuming and tedious job of counting and measuring ommatidia in arthropod eyes. They present two novel open source Python algorithms for automatically quantifying externally and some internally measurable or calculable parameters of the compound eye using 2d and 3d datasets, the latter which is increasingly being generated for anatomical study. They test their algorithms (ODA and ODA-3D) using light micrographs and CT-scanned 3d data sets from previous studies, comparing the results to their own manual measurements and counts. A few insect species were used, varying in size and eye shape and design. They also compare their results to equivalent values in the literature and critically consider the limitations of the automated algorithms.

The algorithms provide a welcomed automation of a tedious data collection process for many compound eye researchers. However, it is not completely free from errors and difficulties associated with the data image quality. Using example data from ant and fruit fly eyes, they show that their ODA gives similar, but not perfectly matching results to manual ommatidia counts and diameter measurements from 2D micrographs, provided the eye is more or less flat. Difficulties would arise when making these measurements from a single image of an eye with a very curved profile.

The second program, the ODA-3D, appears to work reliably in the spherical eyes of two moth species, and to some extent successfully in the vertically elongated oval shaped eyes of a bee, but it was not clear how much manual tweaking and time the program needs for the user to adjust to different, more challenging data. There were issues with varying scan contrast for the bumble bee whereby crystalline cones were unsuccessfully segmented in the dorsal eye, resulting in missing data. This problem may be commonplace with other large data sets and is a difficult one to navigate without manual tweaking. The program appears to work well for good quality scans of eyes with spherical shape. They mention that for large asymmetric eyes, there are still some time consuming stages (e.g. spectral clustering) of the algorithm. Much compound eye research concerns animals with interesting optical specializations, such as acute zones (briefly mentioned). It would be interesting to know how well the ODAs perform with these in future.

Ideally, one would have liked to see a few more examples of the program in practice, although I appreciate that there is not a wealth of 3d data sets available currently. It would be interesting to know if the authors can speculate (or better still, test) whether it work as effectively with animals like crustaceans? These animals tend to possess flat surface corneal lenses, where outward patterns and features (edges of ommatidia) may be less defined in the images? Decapods often have eyes that wrap all the way round an eyestalk – can you speculate on how the ODA-3D will cope with that? I also look forward to seeing if the programs (in future) can be adapted to work with eyes that have irregular ommatidia, lacking a hexagonal lattice distribution, or animals such as shrimp with square facets.

There will be undoubtedly some small errors in the calculations, accumulating from the steps involved, but the programs appear to work well at least for quick* approximations and comparisons between species etc., (*provided the user is able to use Python). The error in measuring the thousands of very small IO angles in non-spherical eyes was particularly apparent in the oval eyes of *B. terrestris*, producing very inaccurate results. The authors have identified this and come up with a solution, to separate them into independent horizontal and vertical components and sample pairs within 15 degree centroids. This reveals interesting IO differences in azimuth / elevation across the eye that seem to have been missed by others previously.

Response: *We agree with the assessment of reviewer three and have spent over a year restructuring the tools and beta testing them with additional users and varied datasets. As mentioned before, we were able to obtain SEM images of *Drosophila melanogaster* and *D. mauritiana* eyes from a collaborator. We ran a battery of tests on the impact of spatial resolution and contrast of the input image on ODA 2D, mentioned above. The battery included SEM images of (1) *D. melanogaster* and (2) *D. mauritiana* eyes taken by Maïke Kittleman, (3) ant eye replicas taken by Ravindra Palavalli-Nettimi, and (4) direct micrographs of *D. melanogaster* eyes. For micro-CT datasets, while we were able to benchmark it on several datasets, most of them were not available publicly and we could not include all the data. We did test a marine decapod eye, which unfortunately did not have sufficient staining to provide clean data, however we did add a feature to search for either hexagonal or square facets. This amounts to searching for*

2 instead of 3 fundamental gratings in the reciprocal image. While we would like to include some of the other features mentioned by reviewer 3, we feel that some of the tests are beyond the scope of the paper describing the method, and as suggested, will be updated in future versions of the program.

Specific comments:

Comment: 1. General = stating that “compound eye being the most common/ prevalent eye design” is repeated many times throughout the manuscript. In the Introduction, the rationale for this study is a little unsatisfying – yes compound eyes are very prevalent, but can you say a little about why is it important to understand them and obtain the data that this work facilitates? Perhaps add a couple of examples of arthropod eye research leading to useful innovations e.g. camera design, pest control traps etc.

Response: *We revised the justification for the introduction and added a few examples of how compound eye research is beneficial and important.*

Comment: 2. Introduction: Another paper to be aware of and reference, which presents a Matlab programme to achieve similar goals: Bagheri et al (2020)
<https://jeb.biologists.org/content/223/1/jeb210195>

Response: *Thank you for the reference. While this is a very impressive program and method, the researchers “manually marked the centre of the corneal facet and the centre of the distal tip of the rhabdom (located at the base of the crystalline cone in fiddler crabs) of each individual ommatidium in microCT reconstructed images (Fig. 1).” (Bagheri et al., 2020) This is precisely the kind of job our program aims to automate. The code was also not publicly available and the program has not been deployed as a usable tool. However, we add a line mentioning papers that do something similar in the introduction.*

Comment: 3. Line 31 = “light level” - change to light intensity and spectral characteristics

Response: *Added suggestion*

Comment: 4. Line 34 = “individual facets, (called ommatidia)” Not strictly true, the facet lenses are the external and outwardly visible part, which contribute to each ommatidium

Response: *Reworded it to be more precise.*

Comment: 5. Line 39 = “many structures” – needs slight rewording, as the only structures that are externally visible are corneal lenses and (sometimes) pseudopupil, which do reveal some insight into optical capabilities in a way camera eyes do not.

Response: *Reworded to be more precise.*

Comment: 6. Line 41 = worth noting that the light reception inside each ommatidial field of view is averaged and does not produce a tiny image.

Response: Added lines “Contrary to popular belief, they generally do not produce a myriad of tiny images on the retina, but average into the functional pixels of the transduced image.”

Comment: 7. Line 55 = “Ommatidia count and diameter can be derived from 2D images” This only works in instances where the eye curvature is slight so that all facets are angled toward the camera/detection chip. There must be difficulties in counting and measuring facets at the eye edge by any means.

Response: While this point is relevant, going further in depth in the introduction may not be as appropriate. Instead, we discuss limitations and advantages of 2D and 3D images for estimation in the discussion. We also changed the language to be less definitive by saying, “Fortunately, ommatidia count and diameter estimation from 2D and 3D eye images can be automated.”

Comment: 8 Lines 77 & 579 = not all superposition eyes are spherical. Some krill and moths present exceptions with asymmetrical shapes. See:
Land, M.F., Burton, F.A. & Meyer-Rochow, V.B. (1979). J. Comp. Physiol. 130, 49.
Warrant E, Bartsch K, G Nther C. (1999). J Exp Biol. 202 (Pt 5):497-511.

Response: Mentioned exceptions and cite them in line 79 and in line 579, we mention it is not always the case that moth eyes are spherical.

Comment: 9. Line 80 = skewed ommatidia, can you clarify a bit more about this or provide an example / reference. It is not clear if you mean the corneal surface lies at a skewed angle from the crystalline cone and/or photoreceptors. It would be beneficial (for those unfamiliar with this phenomena) if you could briefly describe what happens to light entering these facets and how the aperture is effectively reduced, resulting in lower sensitivity.

Response: We have elaborated and clarified what skewed ommatidia are in the text and figure . Ommatidial skewness is examined extensively in a section of Stavenga, (1979), so we also explicitly recommend this chapter for further reading in the text.

Revised text: “Eyes with the longitudinal axes of ommatidia askew to the eye surface are not well approximated by a spherical model. Skewed ommatidia can improve acuity at the expense of field of view (FOV) by pointing more ommatidia onto a small visual field (Figure 1C), or increase FOV at the expense of acuity by spreading a few ommatidia over a large visual field (Figure 1D), but in both cases sacrifice sensitivity by reducing the effective aperture as a function of the skewness angle and refraction (Stavenga 1979). For more information on the optical consequences of ommatidial skewness, see Stavenga (1979:371–77).”

Comment: 10: Line 105 = Specimens and eye imaging

Response: Changed the subsection heading

Comment: Line 106 = what material were the eye molds made from?

Response: *Glue, Added it into methods*

Comment: 12. Lines 114-5 = “l2” subscript needed

Response: *Changed*

Comment: 13. Line 121 = Are these many decimal places accurate / necessary for voxel size?

Response: *Yes, even minor changes to the last few decimal places can throw off the accuracy of the measurements, because that error is compounded over several thousand measurements.*

Comment: 14. Line 132 = voxel size would be worth stating here for the synchrotron data as a comparison to microCT.

Response: *Added the voxel size for the synchrotron dataset as well.*

Comment: 15. Methods = Brief description to say how you made your manual measurements would be good, especially if any software were used to help the process.

Response: *Added a line about manual ommatidial counts.*

Comment: 16. Fig. 3 panels A-D not referenced in legend.

Response: *Referenced A-D in the legend.*

Comment: 17. Line 146 = how are ommatidial diameters calculated and what happens when a facet is missed – e.g. Fig 5a, 4th panel?

Response: *We have elaborated on how these are calculated and what happens when they are missed.*

Revised text: “Optionally, our program can identify these problematic clusters and attempt to correct them using a custom clustering algorithm. Problematic clusters are identified as those with a large proportion ($\geq 10\%$) of elements whose nearest neighbors were elements of a different cluster. For each problematic cluster, the algorithm finds the immediate neighborhood of 6 clusters and applies a nonlinear minimization algorithm (Scipy’s differential evolution) to minimize both 1) the mean projected distance between each point and its cluster center and 2) the aspect ratio of the cluster’s longest side to its shortest in 3D. Because the clusters are approximately parallel on a small scale and elongated along the ommatidial axis, this process often converges on the plane orientation orthogonal to the clusters’ longitudinal axes. The program only incorporates these cluster corrections if they result in substantially less problematic elements ($< 5\%$). This dramatically improves cluster segmentation for non-spherical eyes like the honeybee scan and adds a trivial amount of time to the program’s total duration. Optionally, our program also allows the user to manually edit cluster centers (Supplemental Figure 1 C–D) and then displays the outcome of this segmentation (Supplemental Figure 1 E.)”

Comment: 18.Lines 149-153 = How reliable and time consuming is this step for large or complex eyes, or scans with noise/ poor contrast? Are crystalline cones always contrasting enough in intensity to segment from other eye structures easily across different species and data acquisition modes? Often crystalline cones, being soft and gelatinous in nature, suffer deformations in preserved eyes. Many eyes have accessory pigment cells close by between the CCs that may have similar intensity values in the tomograms. I see you had issues with the dorsal region of the bumblebee eye (line 240), resulting in missing data. Is there any way around this – say to match the facet centres with their underlying photoreceptor column instead?

Response: *We describe the effect of quality and resolution of the scan and add a new figure 5 to quantify this effect. We also include in the discussion our results with scans of poor quality and how to discuss sensitivity of the program to deformations as well as how we found workarounds for some of these issues. The time taken to clean a scan is not an objective metric, instead we discuss which scans require more or less cleaning.*

*Revised Text: “For μ CT, if individual crystalline cones cannot be resolved at each layer, they are likely indiscriminable to the ODA. Further, some species, preparations, and scanning procedures, capture better contrast between crystalline cones and other structures while avoiding structural damage to the specimen. For example, *M. sexta* crystalline cones contrasted sharply with the background scan when prefiltered with just the threshold function. *D. elpenor*, however, had additional noise outside of the eye, and *A. mellifera* had uneven exposure, altering density measurements across locations in the scan and ultimately forcing omission of some data. This may be an unintentional consequence of the contrast enhancing property of a synchrotron light source (Baird and Taylor 2017). Most importantly, ODA-3D erroneously segmented highly skewed ommatidia in the *A. mellifera* scan, resulting in inaccuracies downstream. This could likely be improved by incorporating nonspherical subvolume unfolding, like in Tichit et al. (2022). Preservation techniques can cause small deformations in the eye and while ODA-3D still works minor deformations, scans of highly deformed eyes break certain assumptions about the uniformity of the lattices and cannot be analyzed using this method.”*

Comment: 19. Fig. 4 = scale bar would be better in microns

Response: *We think that while microns could also work, using mm is more useful as it also helps estimate the eye size in units spanning visible distances.*

Comment 20: Line 209-13 = The program appears to create a good approximation of facet lens count and diameter from 2d images, but is not completely reliable/ accurate. Is this due to image contrast differences? Do shadows and reflections pose a problem for recognising the facets?

Response: *We don't know if they do, but we suspect it's more related to the reduced projected area of the ommatidia due to the curvature of the eye. We discuss some of the reasons for why the ODA and ODA-3D may be inaccurate or might not work. We also mention that more testing is required.*

Comment: 21. In some cases where microscope images of intact eyes are used, miscalculations might also arise because of how the 2d images represent the ommatidia at the borders of the eye, due to their side-on angle with respect to the camera. Manual counts would also be erroneous for the same reason. Fig. 5c shows that lens centres appear to cluster together at the edges due to the eye curve. Is this affecting your facet diameter calculations?

Response: *We now explicitly state how we calculate ommatidial diameter for manual counting. See revised section in "Data Validation and Manual counts"*

Comment 22. Fig. 5A. Facets missed by the program appear obvious to the human eye due to pattern recognition, could this be incorporated into the algorithm somehow to prevent under-sampling?

Response: *This is a great point. We developed a GUI that allows users to manually delete, move, and add coordinates for ommatidia. This is still in testing but is an optional stage of the ODA and is shown in Supplemental Fig 1 C and D.*

Comment 23. Fig. 5B. It would help to have an x=y line or grid background across the figures to facilitate visualization of how many points lie above / below the lines for comparison.

Response: *Added X=Y line*

Comment: 24. Line 226 = "Facets actually curve slightly with the curvature of the eye" Yes, for light micrograph images of intact eyes, only facet diameters from the central region of the eye facing toward the camera can be accurately measured. How were manual measurements made? Lenses tend to become smaller at the edges of an eye, where their profile will be side-on and difficult to measure in a single 2d image by any means.

Response: *We added a section in the methods specifying how we obtained the manual counts and ommatidial parameters.*

Comment: 25. 235 = "The 2D histograms and the superimposed ommatidial centers suggested that the first count was more accurate for *M. sexta* while the second count was more accurate for *D. elpenor*." - I was slightly confused here by your comparisons of two sets of counts, as it was not mentioned previously that you ran the program on the same data twice, if that is what you meant?

Response: *The same dataset was not run twice. Instead, we previously used a count before and after removing some points due to the analysis. This is no longer an issue, and this text has been removed since analyzing the data with the newer versions of ODA.*

Comment: 26. Lines 242-250: Comparisons with literature values. Ommatidia counts will vary naturally across individuals of a species, so this isn't a strong evidence for accuracy. It doesn't compare direct measurements so could perhaps be more of a discussion point rather than presented in the results.

Response: *In the revised text, we refrain from claiming it is a measure of accuracy but simply mention how much it varies with existing measurements, which is useful information to include. We think it fits better in the results than the discussion because these are not interpretations but a numerical comparison with secondary sources of data.*

Comment: 27. Lines 268- “However, instead of using the center of the head as our center, we used the center found in step B of ODA-3D, which is near but not exactly the center of the head.” Why was an off-center point used and does this apply to all eyes processed this way? Does it perhaps approximate to the centre of the basal membrane or something? Also, the sentence is a bit clunky to read with the word “center” used 4 times.

Response: *We altered the sentence to be clearer and less confusing:*

Revised text: “We used the center from step B of ODA-3D, which is near the center of the head and chose a radius of 10 cm based on visual fixation behavior (Wehner and Flatt 1977)”

Comment: 29. Lines 297 and 302 = Figure B?

Response: *Included correct figure reference*

Comment: 30. Line 299 = These comparisons to measurements by Baumgärtner (1929) seem a little bit tenuous, due to lack of species information. Are there any (recent!) sources of data with known bee species you can compare to?

Response: *We included comparisons with a more recent source, Taylor et al. (2018), which took measurements on this exact scan.*

Comment 31. Line 302 = reword slightly so that it is clear from the start that the “previous measurements” refer to values published in another study (+give ref). In general it might be worth replacing “previous” with the name of the authors to avoid confusing with your own previous measurements.

Response: *We revised text to include citations and make the wording less ambiguous.*

Comment: 32. Fig. 6 & 7 = include animal species in the legends

Response: *We now include species names in the figure legends.*

33. Fig. 7b = in the table, top row, are the two count values separated by – symbol a range or two separate values that you calculated from running the programs twice? The range in values seems very high, many hundreds of ommatidia different.

Response: *We have changed this in the latest version, which provides better approximations and without this ambiguity, and now only a single value is included.*

Comment: 34. Line 332 = End of Results could perhaps be reorganised “Similar tradeoffs” – perhaps this is referring to the skewness rather than fixation and binocular FOV just stated. Worth adjusting for readability.

Response: *We have revised the results to improve readability*

Comment: 35. Line 364 = “As far as we know, this has not been measured previously, though it is consistent with previous measurements...” could do with re-wording!

Response: *We revised the text to be more readable*

36. Line 392 = Add in that there may also be glial pigment cell interactions between the crystalline cones or deformation of the soft cells, which may present difficulties in auto segmentation in this region of the eye.

Response: *We have included suggestions on results.*

Figure Revisions

In the process of responding to the reviewer comments, we revised all of the results figures. Here we specify the changes we made to each.

Figure 4: Since we changed to using the *Apis mellifera* scan, we updated the image and data plotted using the latest version of the program, but essentially demonstrate the same as before.

Figure 5: A) We removed one of the 6 ant eye images because it was not included in the benchmarking results. We also added the original images for easier visual comparison. B) We added the benchmark results to 5B, C) changed the colormap to match 5A, and D) added the $x=y$ line based on the reviewer comments.

Figure 6: A) We added a fruit fly scan to the left and replaced the results for *B. terrestris* with *A. mellifera*. B) We added plots of the manual and ODA durations for each scan. D) this is a new plot demonstrating the projection differences between a spherical and non-spherical eye. E) We added the *D. elpenor* IO angle results to again compare a spherical and non-spherical eye.

Supplemental Figure 1: we added this figure to demonstrate key graphical features that we added to the software that allow for checking the performance at each stage of the ODA-3D pipeline. A) The prefiltering stage allows for selecting a subset of voxel density values. B) After fitting the surface to the points, you can preview the residual distance of each point. C-D) After finding ommatidia, the user can manually add, delete, or clear all output to correct for errors in the ODA. E) Once the volume has been segmented, you can preview the labeling of the segmentation stage. F) Finally, after all the measurements have been taken, you can inspect all of the outcome parameters and each stage of the pipeline using a 3D interactive interface.

** See the Nature Portfolio author and referees' website at www.nature.com/authors for information about policies, services and author benefits

Communications Biology is committed to improving transparency in authorship. As part of our efforts in this direction, we are now requesting that all authors identified as 'corresponding author' create and link their Open Researcher and Contributor Identifier (ORCID) with their account on the Manuscript Tracking System prior to acceptance. ORCID helps the scientific community achieve unambiguous attribution of all scholarly contributions. You can create and link your ORCID from the home page of the Manuscript Tracking System by clicking on 'Modify my Springer Nature account' and following the instructions in the link below. Please also inform all co-authors that they can add their ORCID to their accounts and that they must do so prior to acceptance.

If you experience problems in linking your ORCID, please contact the Platform Support Helpdesk.

Our flexible approach during the COVID-19 pandemic

If you need more time at any stage of the peer-review process, please do let us know. While our systems will continue to remind you of the original timelines, we aim to be as flexible as possible during the current pandemic.

COMMSBIO - This email has been sent through the Springer Nature Tracking System NY-610A-NPG&MTS

Confidentiality Statement:

This e-mail is confidential and subject to copyright. Any unauthorised use or disclosure of its contents is prohibited. If you have received this email in error please notify our Manuscript Tracking System Helpdesk team at <http://platformsupport.nature.com> .

Details of the confidentiality and pre-publicity policy may be found here <http://www.nature.com/authors/policies/confidentiality.html>

Privacy Policy | Update Profile

DISCLAIMER: This e-mail is confidential and should not be used by anyone who is not the original intended recipient. If you have received this e-mail in error please inform the sender and delete it from your mailbox or any other storage mechanism. Springer Nature Limited does not accept liability for any statements made which are clearly the sender's own and not expressly made on behalf of Springer Nature Ltd or one of their agents.

Please note that Springer Nature Limited and their agents and affiliates do not accept any responsibility for viruses or malware that may be contained in this e-mail or its attachments and it is your responsibility to scan the e-mail and attachments (if any).

Springer Nature Ltd. Registered office: The Campus, 4 Crinan Street, London, N1 9XW. Registered Number: 00785998 England

REVIEWERS' COMMENTS:

Reviewer #1 (Remarks to the Author):

I have read the revised manuscript COMMSBIO-21-0309-T and confirms the authors have addressed properly my previous concerns.

Thus, I support the publication of the new version by Commun. Biol.

Reviewer #3 (Remarks to the Author):

Thank you for your responses. I am satisfied with the revisions and that you have carefully considered every point and made both great and small improvements to this work.